# The C-terminal tail of the bacterial translocation ATPase SecA modulates its activity

Mohammed Jamshad[1†], Timothy J Knowles[1†], Scott A White[1], Douglas G Ward[2], Fiyaz Mohammed[3], Kazi Fahmida Rahman[1], Max Wynne[1], Gareth W Hughes[1], Günter Kramer[4], Bernd Bukau[4], Damon Huber[1]*

[1]Institute for Microbiology and Infection, School of Biosciences, University of Birmingham, Birmingham, United Kingdom; [2]Institute of Cancer and Genomic Sciences, University of Birmingham, Birmingham, United Kingdom; [3]Institute of Immunology and Immunotherapy, University of Birmingham, Birmingham, United Kingdom; [4]Center for Molecular Biology of Heidelberg University (ZMBH), German Cancer Research Center (DKFZ), ZMBH-DKFZ Alliance, Heidelberg, Germany

**\*For correspondence:**
d.huber@bham.ac.uk

[†]These authors contributed equally to this work

**Competing interests:** The authors declare that no competing interests exist.

**Abstract** In bacteria, the translocation of proteins across the cytoplasmic membrane by the Sec machinery requires the ATPase SecA. SecA binds ribosomes and recognises nascent substrate proteins, but the molecular mechanism of nascent substrate recognition is unknown. We investigated the role of the C-terminal tail (CTT) of SecA in nascent polypeptide recognition. The CTT consists of a flexible linker (FLD) and a small metal-binding domain (MBD). Phylogenetic analysis and ribosome binding experiments indicated that the MBD interacts with 70S ribosomes. Disruption of the MBD only or the entire CTT had opposing effects on ribosome binding, substrate-protein binding, ATPase activity and in vivo function, suggesting that the CTT influences the conformation of SecA. Site-specific crosslinking indicated that F399 in SecA contacts ribosomal protein uL29, and binding to nascent chains disrupts this interaction. Structural studies provided insight into the CTT-mediated conformational changes in SecA. Our results suggest a mechanism for nascent substrate protein recognition.
DOI: https://doi.org/10.7554/eLife.48385.001

## Introduction

In *Escherichia coli*, approximately a quarter of all newly synthesised proteins are transported across the cytoplasmic membrane by the Sec machinery (*Cranford-Smith and Huber, 2018*). Of these, the majority (~65%) require the activity of SecA for translocation across the membrane. SecA is an evolutionarily conserved and essential ATPase that is required for protein translocation in bacteria (*Cranford-Smith and Huber, 2018*). The catalytic core of SecA (amino acids ~1–832 in *E. coli*) contains five domains (*Figure 1—figure supplement 1*): nucleotide binding domain-1 (NBD1; amino acids 9–220 and 378–411), the polypeptide crosslinking domain (PPXD; 221–377), nucleotide binding domain-2 (NBD2; 412–620), the α-helical scaffold domain (HSD; 621–672 and 756–832) and the α-helical wing domain (HWD; 673–755). Binding and hydrolysis of ATP at the NBD1/NBD2 interface cause conformational changes in the HSD and HWD, which are required for the translocation of substrate proteins (*Cranford-Smith and Huber, 2018*; *Collinson et al., 2015*). In addition, the PPXD undergoes a large conformational change, swinging from a position near the HWD (the 'closed' conformation) to a position near NBD2 (the 'open' conformation) (*Zimmer et al., 2008*; *Zimmer and Rapoport, 2009*; *Chen et al., 2015*). SecA binds to substrate protein in the groove formed between

the PPXD and the two NBDs, and the PPXD serves as a 'clamp' that locks unfolded substrate proteins into this groove when it is in the open conformation (*Zimmer and Rapoport, 2009*).

SecA can recognise its substrate proteins cotranslationally. SecA binds to the ribosome (*Huber et al., 2011*), and ribosome binding facilitates the recognition of nascent substrate proteins (*Huber et al., 2017*; *Huber et al., 2011*). The binding site for SecA on the ribosome includes ribosomal protein uL23, which is located adjacent to the opening of the polypeptide exit channel (*Huber et al., 2011*). Structural and biochemical studies indicate that ribosome binding is mediated by two regions in the catalytic core: the N-terminal α-helix (*Singh et al., 2014*) and the N-terminal portion of the HSD (*Huber et al., 2011*) (*Figure 1—figure supplement 1*). The structure of the SecA-ribosome complex was recently determined at medium resolution (~11 Å) by cryo-electron microscopy (*Singh et al., 2014*). However, the molecular mechanism governing the recognition of nascent substrate proteins is unknown.

In addition to the catalytic core, most SecA proteins contain a relatively long C-terminal tail (CTT; also known as the C-terminal linker [*Hunt et al., 2002*]), whose function is not well understood (*Figure 1A*). In *E. coli*, the CTT (833-901) contains of a small metal binding domain (MBD; 878–901) and a structurally flexible linker domain (FLD; 833–877). Although it is not required for protein translocation (*Or et al., 2002*; *Or et al., 2005*; *Fekkes et al., 1997*), *E. coli* strains producing a C-terminally truncated SecA protein display modest translocation defects (*Or et al., 2005*; *Grabowicz et al., 2013*). The MBD coordinates a metal ion (thought to be $Zn^{2+}$) *via* a conserved $CXCX_8C(H/C)$ motif (*Dempsey et al., 2004*; *Fekkes et al., 1999*). In *E. coli*, the MBD is required for interaction of SecA with SecB (*Kimsey et al., 1995*; *Breukink et al., 1995*; *Fekkes et al., 1997*), a molecular chaperone that is required for the secretion of a subset of Sec substrate proteins (*Randall and Hardy, 2002*). Although not all SecA proteins contain an MBD, the MBD is conserved in many species that do not have a SecB homologue (*Dempsey et al., 2004*).

In this study, we investigated the role of the CTT in the recognition of nascent substrate proteins by SecA. Phylogenetic and sequence analysis of the CTT suggested that it could be involved in binding of SecA to ribosomes, which we confirmed using ribosome cosedimentation and chemical crosslinking approaches. Strikingly, disruption of the MBD alone or the entire CTT had opposing effects on multiple activities of SecA, suggesting that the CTT affects conformation of the catalytic core. Mass spectrometry, x-ray crystallography, and small-angle x-ray scattering experiments indicated that the FLD is bound in the substrate binding groove and affects the conformation of the PPXD. Finally, site specific chemical crosslinking suggested that binding of the MBD to the ribosome allows full-length SecA to interact with nascent substrate proteins. Taken together, our results provide insight into the molecular mechanism underlying nascent substrate recognition by SecA.

## Results

### Evolutionary distribution of the MBD of SecA

To investigate the evolutionary distribution of the CTT, we analysed the sequences of 156 SecA proteins from bacterial species in 155 phylogenetic families using ClustalOmega (*McWilliam et al., 2013*). The phylogenetic tree produced by this analysis generally placed SecA proteins from more closely related species (*e.g.* those in the same phylogenetic class) into similar groups (*Figure 1B*; *Supplementary files 1* and *2*). The majority of SecA proteins (143) contained a CTT (*Figure 1B*, red and black branches). Of these, 117 contained an MBD (*Figure 1B*, black branches). A small minority (13) lacked the CTT entirely (*Figure 1B*, yellow branches). Of the 69 SecA proteins from species that contained a SecB homologue (*Figure 1B*, starred species), only two lacked an MBD. The strong co-conservation of the MBD and SecB suggests that there is strong selective pressure to maintain the MBD in species possessing SecB, consistent with previous studies indicating that the MBD is required for binding of SecA to SecB (Fekkes, *Fekkes et al., 1997*; *Fekkes et al., 1999*; *Zhou and Xu, 2003*; *Randall et al., 2004*). However, a significant number of species that lack SecB (52) also contain a SecA protein with an MBD. Furthermore, many of the residues implicated in SecB binding were strongly conserved in these MBDs (*Figure 1C*, arrowheads) (*Zhou and Xu, 2003*). These results suggested that the MBD has an evolutionarily conserved function in addition to its role in binding to SecB.

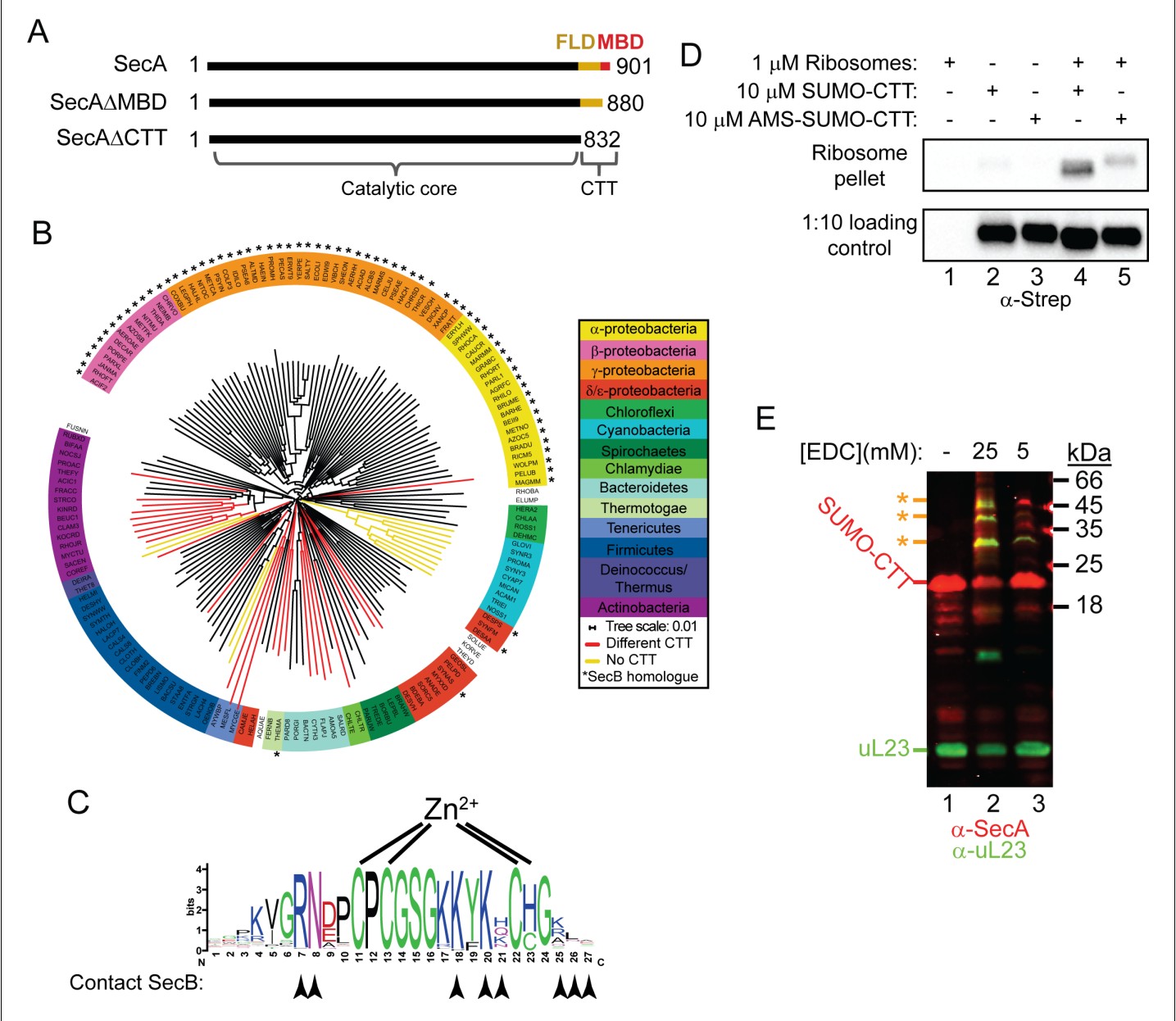

**Figure 1.** Phylogenetic analysis of the CTT and binding of *E. coli* CTT to the ribosome. (**A**) Schematic diagram of the primary structure of SecA, SecAΔMBD and SecAΔCTT. Structures are oriented with the N-termini to the left, and the amino acid positions of the N- and C-termini are indicated. Residues of the catalytic core and the CTT are indicated below. Catalytic core, black; FLD, yellow; MBD, red. (**B**) Phylogenetic tree of the SecA proteins of 156 representative species from 155 different bacterial families. Species names are given as the five-letter organism mnemonic in UniProtKB (***The UniProt Consortium, 2017***). Taxonimical classes are colour-coded according to the legend. Leaves representing SecA proteins with an MBD are coloured black. Those with CTTs lacking a MBD are coloured red, and those that lack a CTT entirely are coloured yellow. Species that also contain a SecB protein are indicated with a star (*). (**C**) Logo of the consensus sequence of the MBD generated from the 117 species containing the MBD in the phylogenetic analysis. Positions of the metal-coordinating amino acids are indicated above. Amino acids that contact SecB in the structure of the MBD-SecB complex (***Zhou and Xu, 2003***) (1OZB) are indicated by arrowheads below. (**D**) Binding reactions containing 1 µM ribosomes, 10 µM SUMO-CTT and 10 µM AMS-modified SUMO-CTT (AMS-SUMO-CTT) were equilibrated at room temperature and layered on a 30% sucrose cushion. Ribosomes were then sedimented through the cushion by ultracentrifugation. Samples were resolved on SDS-PAGE and probed by western blotting against the Strep tag using HRP-coupled Streptactin. (**E**) 10 µM SUMO-CTT containing an N-terminal Strep(II)-tag was incubated with 1 µM purified ribosomes and treated with 5 mM or 25 mM EDC, as indicated. Samples were resolved by SDS-PAGE and analysed by western blotting by simultaneously probing against SecA (red) and ribosomal protein uL23 (green). The positions of SUMO-CTT, L23 and crosslinking adducts between them (*) are indicated at left.

DOI: https://doi.org/10.7554/eLife.48385.002

*Figure 1 continued on next page*

*Figure 1 continued*

The following source data and figure supplements are available for figure 1:

**Source data 1.** Clustal Omega alignment of SecA proteins used to construct phylogenetic tree in *Figure 1*.
DOI: https://doi.org/10.7554/eLife.48385.005
**Source data 2.** Phylogenetic tree data generated by Clustal Omega used to construct *Figure 1B and C*.
DOI: https://doi.org/10.7554/eLife.48385.006
**Figure supplement 1.** Structural model of the catalytic core of SecA in the 'closed' conformation.
DOI: https://doi.org/10.7554/eLife.48385.003
**Figure supplement 2.** SUMO-MBD cosediments with ribosomes.
DOI: https://doi.org/10.7554/eLife.48385.004

## Binding of the CTT to the ribosome

Many of the most highly conserved residues in the MBD consensus sequence (including in species that lack SecB) are positively charged and surface exposed (*Figure 1C*), which suggested that the MBD could also bind to the negatively charged surface of the ribosome. A fusion between the small ubiquitin-like modifier (SUMO) from *Saccharomyces cerevisiae* and the CTT of *E. coli* SecA (SUMO-CTT) co-sedimented with ribosomes through a sucrose cushion during ultracentrifugation, indicating that the CTT binds to ribosomes (*Figure 1D*, lanes 2 and 4). A shorter protein fusion containing only the MBD (SUMO-MBD) also cosedimented with ribosomes (*Figure 1—figure supplement 2*), indicating that the MBD is responsible for this ribosome binding activity, and modification of the metal-coordinating cysteines with AMS disrupted the ability of SUMO-CTT to cosediment with ribosomes (*Figure 1D*, lanes 3 and 5). Incubation of SUMO-CTT with ribosomes in the presence of 5 mM and 25 mM EDC (a non-specific crosslinking agent) resulted in the appearance of several crosslinking products. These products cross-reacted with antibodies against uL23 and SecA (*Figure 1E*), suggesting that the CTT binds in the vicinity of the opening of the polypeptide exit tunnel similar to full-length SecA.

## Effect of C-terminal truncations on the affinity of SecA for ribosomes

We next determined the affinity of C-terminal truncation variants of SecA for the ribosome using fluorescence anisotropy (*Huber et al., 2011*) (*Figure 2A* and *Table 1*). The equilibrium dissociation constant ($K_D$) of the complex between full-length SecA and ribosomes was ~640 nM, similar to previously published figures (*Figure 2A* and *Table 1*) (*Huber et al., 2011*). Truncation of the C-terminal 69 amino acids of SecA (SecAΔCTT) caused a modest, but reproducible, increase in the $K_D$ of the SecA-ribosome complex (920 nM) (*Figure 2A* and *Table 1*). However, truncation of the C-terminal 21 amino acids (SecAΔMBD) significantly increased the affinity of SecA for the ribosome ($K_D$ = 160 nM) (*Figure 2A* and *Table 1*). These differences in affinity were sufficient to affect the amount of SecA that cosedimented with ribosomes during ultracentrifugation (*Figure 2B*, lanes 4–6).

## Effect of truncations on affinity of SecA for nascent polypeptides

To investigate whether the truncations affected the affinity of SecA for nascent chains, we examined binding of SecAΔMBD and SecAΔCTT to ribosome nascent chain complexes (RNCs) containing arrested nascent SecM. SecM is a model nascent SecA substrate protein (*Huber et al., 2017*; *Huber et al., 2011*). Similar to full-length SecA (*Huber et al., 2011*), binding of SecAΔMBD and SecAΔCTT to non-translating ribosomes was sensitive to high concentrations of salt in the binding buffer (*Figure 2B*, lanes 7–9). Binding of SecA and SecAΔCTT to ribosomes in the presence of 250 mM potassium acetate was stabilised by the presence of arrested nascent SecM (*Figure 2B*, lanes 10 and 12). However, nascent SecM did not stabilise binding of SecAΔMBD to RNCs under the same conditions (*Figure 2B*, lane 11). These results suggested that SecAΔMBD is defective for binding to nascent substrate protein.

## Site-specific crosslinking of SecA to ribosomes

To investigate binding of SecA to the ribosome in more detail, we incorporated *p*-benzoyl-L-phenylalanine (Bpa) into SecA at positions 56, 260, 299, 399, 406, 625, 647, 665, 685, 695, 748 and 796 using nonsense suppression (*Figure 3A and B*) (*Singh et al., 2014*; *Huber et al., 2011*;

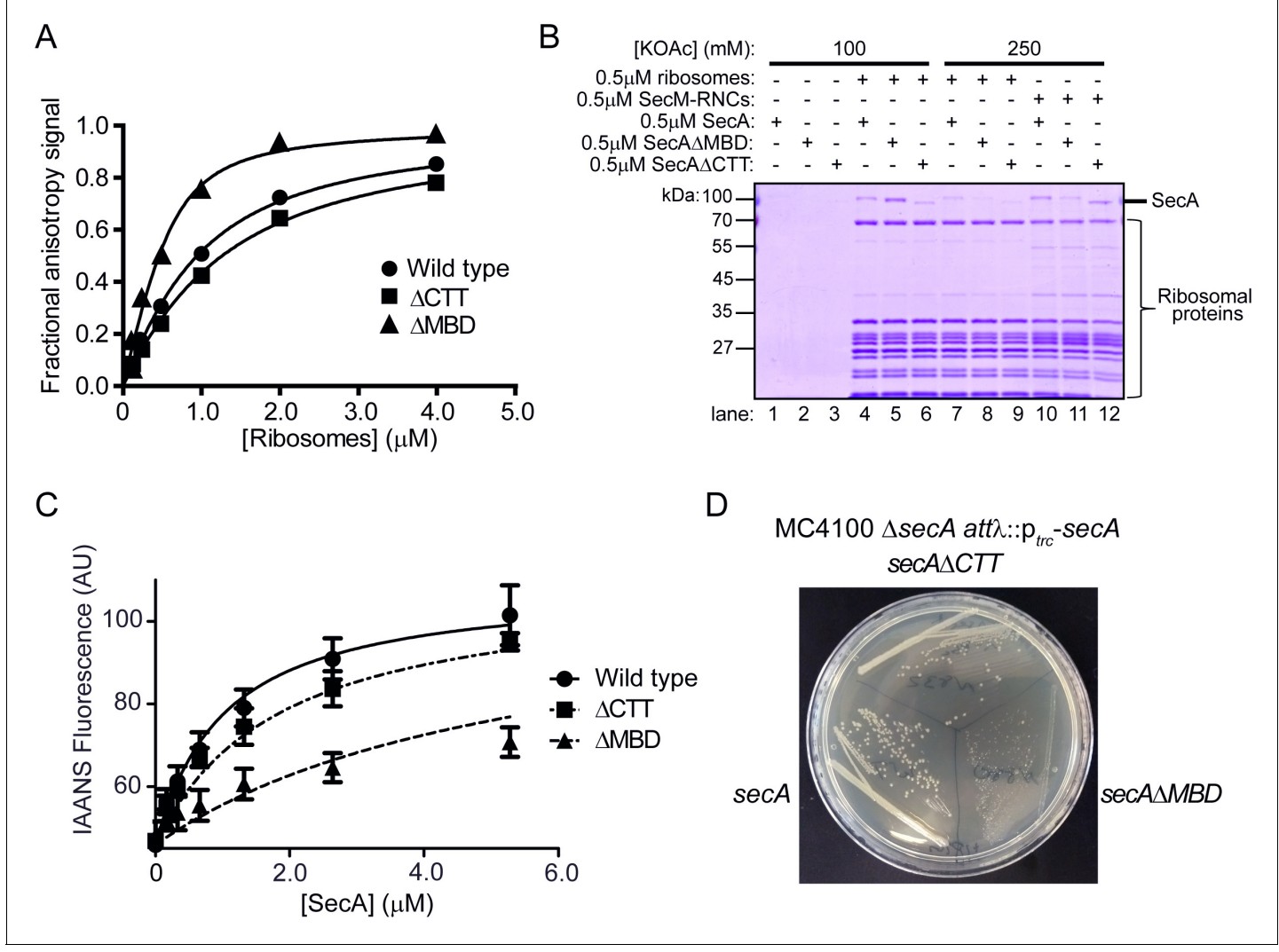

**Figure 2.** Effect of C-terminal truncations on SecA function in vitro and in vivo. (**A**) 900 nM Ru(bpy)$_2$(dcbpy)-labelled SecA (Wild type; circles), SecAΔMBD (ΔMBD; triangles) or SecAΔCTT (ΔCTT; squares) was incubated in the presence of increasing concentrations of purified 70S ribosomes. Because error bars corresponding to one standard deviation obscured the symbols, they were omitted from the graph. The equilibrium dissociation constant ($K_D$) of the complex was determined by fitting the increase in fluorescence anisotropy from the Ru(bpy)$_2$(dcbpy) (lines; *Table 1*). (**B**) 0.5 μM SecA, SecAΔMBD or SecAΔCTT was incubated in the absence (lanes 1–3) of ribosomes, in the presence of 0.5 μM vacant 70S ribosomes (lanes 4–9) or in the presence of 0.5 μM RNCs containing nascent SecM peptide (lanes 10–12). Where indicated, binding reactions were incubated in the presence of 100 mM (lanes 1–6) or 250 mM (lanes 7–12) potassium acetate (KOAc). Binding reactions were layered on a 30% sucrose cushion and ribosomes were sedimented through the sucrose cushion by ultracentrifugation. Ribosomal pellets were resolved by SDS-PAGE and stained by Coomassie. (**C**) 600 nM IAANS-VipB peptide was incubated with increasing concentrations of SecA (Wild type; circles), SecAΔMBD (ΔMBD; triangles) or SecAΔCTT (ΔCTT; squares). Confidence intervals represent one standard deviation. The $K_D$ for the SecA-peptide complex was determined by fitting the increase in IAANS fluorescence upon binding to SecA (lines; *Table 1*). (**D**) Growth of strains producing SecA (DRH1119; bottom left), SecAΔMBD (DRH1120; bottom right) and SecAΔCTT (DRH1121; top) on LB plates containing 100 μM IPTG.

DOI: https://doi.org/10.7554/eLife.48385.007

The following figure supplements are available for figure 2:

**Figure supplement 1.** CD spectra of SecA, SecAΔMBD and SecAΔCTT.

DOI: https://doi.org/10.7554/eLife.48385.008

**Figure supplement 2.** Thermal denaturation plots of SecA, SecAΔMBD and SecAΔCTT.

DOI: https://doi.org/10.7554/eLife.48385.009

**Figure supplement 3.** Expression of SecA, SecAΔMBD and SecAΔCTT in strains DRH1119, DRH1120 and DRH1121.

DOI: https://doi.org/10.7554/eLife.48385.010

**Table 1.** Biochemical properties of wild-type and mutant SecA proteins.

| SecA variant | $K_D$ Ribosomes[*] | $K_D$ VipB[†] | Basal ATPase activity[‡] | $T_M$[§] |
|---|---|---|---|---|
| Wild type | 640 ± 33 nM | 0.9 μM | 0.053 ± 0.02 s$^{-1}$ | 40.7 ± 0.09°C |
| SecAΔMBD | 160 ± 35 nM | 1.7 μM | <0.001 s$^{-1}$ | 42.0 ± 0.08°C |
| SecAΔCTT | 920 ± 38 nM | 5.9 μM | 0.91 ± 0.02 s$^{-1}$ | 40.0 ± 0.1°C |
| SecA$^{C885A/C887A}$ | ND[¶] | >10 μM | <0.001 s$^{-1}$ | ND[¶] |
| SecA$^{Bpa852}$ | ND[¶] | >10 μM | <0.001 s$^{-1}$ | ND[¶] |

[*]Equilibrium dissociation constant of the complex between SecA and non-translating 70S ribosomes as determined by fluorescence anisotropy. Confidence intervals are the standard error of the fit.
[†]Equilibrium dissociation constant of the complex between SecA and IAANS-labelled VipB peptide as determined by change in fluorescence.
[‡]Rate of ATP hydrolysis by SecA in the absence of substrate protein and SecYEG
[§]Denaturation midpoint temperature as determined by the change in circular dichroism at 222 nm.
[¶]not determined
DOI: https://doi.org/10.7554/eLife.48385.011

*Chin et al., 2002*). The side chains of the amino acids at these positions are located on the surface of SecA that binds to the ribosome (*Figure 3A*). Bpa contains a photoactivatable side chain that forms covalent crosslinks to nearby molecules containing C-H bonds. In the presence of purified 70S ribosomes, SecA containing Bpa at positions 399 (SecA$^{Bpa399}$) and 406 (SecA$^{Bpa406}$) produced additional high molecular weight bands in SDS-PAGE, which were recognised by α-SecA antiserum (*Figure 3C*). Analysis of the high-molecular weight band produced by SecA$^{Bpa399}$ by mass-spectrometry (LC-MS/MS) indicated that it was an adduct between SecA and ribosomal protein uL29. uL29 is located adjacent to uL23 on the ribosomal surface, and both F399 and K406 appear to contact uL29 in the structure of the SecA-ribosome complex determined by cryo-electron microscopy (*Figure 3A*) (*Singh et al., 2014*). SecA containing Bpa at position 299 also produced a crosslinking adduct that migrated with a larger apparent molecular weight than the SecA-uL29 adduct (*Figure 3C*). However, the identity of the crosslinking protein is unknown. SecAΔMBD$^{Bpa399}$ also produced a high molecular weight crosslinking adduct in the presence of ribosomes, and LC-MS/MS confirmed the presence of both SecA and uL29 in the band (*Figure 3D*), indicating SecAΔMBD binds to ribosomes at the same site as full-length SecA.

## Crosslinking of SecA and SecAΔMBD to RNCs

To investigate the effect of a nascent chain on binding of SecA to the ribosome, we incubated SecA$^{Bpa399}$ with RNCs containing arrested nascent SecM or maltose binding protein (MBP) (*Figure 3D* and *Figure 3—figure supplement 1*). The presence of a nascent chain long enough to interact with SecA inhibited crosslinking of SecA$^{Bpa399}$ to uL29 (*Huber et al., 2017*), but the presence of an arrested nascent chain that is too short to interact with SecA did not significantly affect crosslinking to uL29 (*Figure 3—figure supplement 2*). These results suggest that binding to nascent polypeptide causes a conformational change in SecA, which affects its interaction with the ribosome. In contrast, the presence of nascent substrate protein did not affect crosslinking of SecAΔMBD$^{Bpa399}$ to uL29 (*Figure 3D*), consistent with the inability of SecAΔMBD to bind to nascent chains.

## Effect of truncations on affinity for free polypeptides

We next examined the affinity of SecA, SecAΔMBD and SecAΔCTT for free polypeptide. To this end, we determined the affinity of SecA for a short peptide, VipB, which was labelled with an environmentally sensitive fluorophore (IAANS-VipB; *Pietrosiuk et al., 2011*) that produces an increase in fluorescence upon binding to SecA. The affinities of SecA and SecAΔCTT for IAANS-VipB ($K_D$ = 0.9 μM and 1.7 μM, respectively) were consistent with the previously reported affinity of SecA for unfolded substrate protein (*Gouridis et al., 2009*) (*Figure 2C* and *Table 1*). However, the affinity of SecAΔMBD for IAANS-VipB was significantly lower ($K_D$ = 5.9 μM). Furthermore, alanine substitutions in two of the metal-coordinating cysteines (SecA$^{C885A/C887A}$) greatly reduced the affinity of SecA for IAANS-

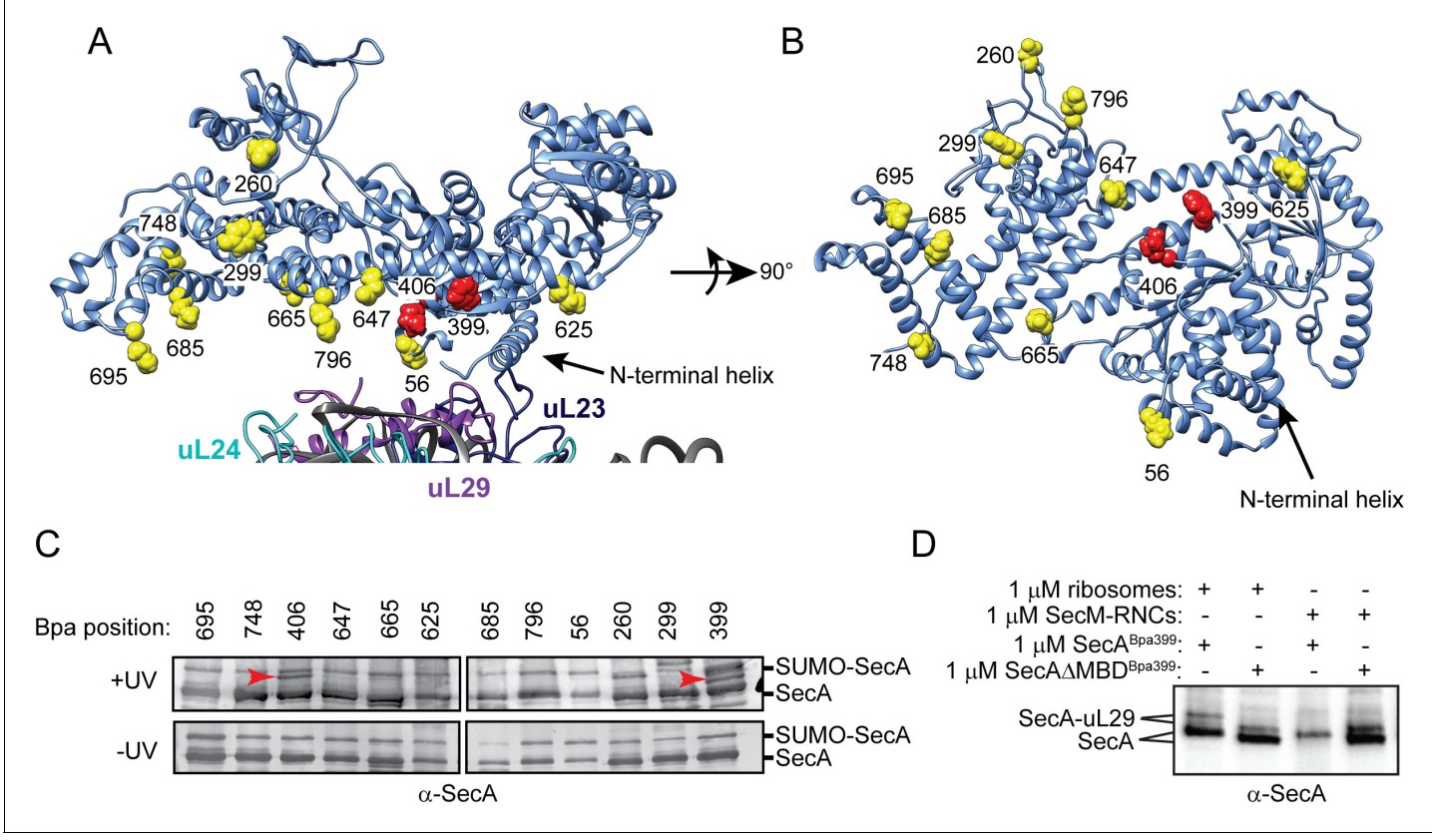

**Figure 3.** Site-specific crosslinking of SecA to purified ribosomes and ribosome-nascent chain complexes. (**A and B**) Sites of incorporation of Bpa in the structure of *E. coli* SecA. (**A**) Fit of the high resolution structure of SecA (PDB code 2VDA [*Gelis et al., 2007*]) and the 70S ribosome (PDB code 4V4Q [*Schuwirth et al., 2005*]) to the cryoEM structure of the SecA ribosome complex (EMD-2565 [*Singh et al., 2014*]). (**B**) View of SecA from the ribosome-interaction surface. Amino acid positions where Bpa was incorporated are represented in space fill (yellow). Positions that crosslink to ribosomal proteins are coloured red. The locations of the N-terminal α-helix of SecA and of ribosomal proteins uL23 (dark blue), uL29 (purple) and uL24 (cyan) are indicated. Structural models were rendered using Chimera v. 1.12 (*Pettersen et al., 2004*). (**C**) Bpa-mediated photocrosslinking of SecA variants to vacant 70S ribosomes. 1 μM purified ribosomes were incubated with 1 μM SecA containing BpA at the indicated position and exposed to light at 365 nm (above) or incubated in the dark. Crosslinking adducts consistent with the molecular weight of a covalent crosslink to ribosomal proteins are indicated with red arrowheads. The positions of full-length SecA and uncleaved SUMO-SecA protein are indicated to the right. (**D**) 1 μM SecA$^{Bpa399}$ or SecAΔMBD$^{Bpa399}$ was incubated with 1 μM non-translating 70S ribosomes or 1 μM arrested RNCs containing nascent SecM (SecM-RNCs) and exposed to light at 365 nm. The positions of full-length SecA and the SecA-uL29 crosslinking adduct are indicated. In (**C and D**), samples were resolved using SDS-PAGE and probed by western blotting using anti-SecA antiserum. LC-MS/MS analysis of the high-molecular weight bands produced by SecA$^{Bpa399}$ and SecAΔMBD$^{Bpa399}$ in the presence of vacant 70S ribosomes indicated that they contained both SecA and ribosomal protein uL29.

DOI: https://doi.org/10.7554/eLife.48385.012

The following figure supplements are available for figure 3:

**Figure supplement 1.** Crosslinking of SecA$^{Bpa399}$ to RNCs containing arrested nascent full-length SecM and MBP.

DOI: https://doi.org/10.7554/eLife.48385.013

**Figure supplement 2.** Crosslinking of SecA$^{Bpa399}$ to RNCs containing arrested nascent chains with different lengths.

DOI: https://doi.org/10.7554/eLife.48385.014

VipB (*Table 1*), suggesting that disrupting the structure of the MBD was sufficient to cause this decrease in affinity.

## Effect of truncations on the ATPase activity of SecA

To investigate the effect of the truncations on the ATPase activity of SecA, we determined the basal ATPase rates of SecA, SecAΔMBD, SecAΔCTT and SecA$^{C885A/C887A}$. The ATP turnover rate for full-length SecA was 0.05 s$^{-1}$ (*Table 1*), consistent with previously reported figures (*Huber et al., 2011*). Deletion of the entire CTT caused a > 10 fold increase in the basal ATPase activity compared to the full-length protein (0.9 s$^{-1}$) (*Table 1*), suggesting that the FLD inhibits the ATPase activity of SecA.

SecA$\Delta$MBD and SecA$^{C885A/C887A}$ did not hydrolyse ATP at a detectable rate, suggesting that the MBD is required to relieve the FLD-mediated autoinhibition.

## Effect of truncations on the folding of SecA

We next investigated the effect of the C-terminal truncations on the secondary structure content and thermal stability of SecA using circular dichroism (CD) spectroscopy. The CD spectra of the three proteins indicated that they were fully folded (*Figure 2—figure supplement 1*). However, the denaturation midpoint temperature ($T_M$) of SecA$\Delta$MBD (42°C) was ~1.5°C higher relative to that of the full-length protein and ~2°C higher than that of SecA$\Delta$CTT (*Figure 2—figure supplement 2*), suggesting that SecA$\Delta$MBD was more stably folded than SecA or SecA$\Delta$CTT.

## SecA truncation variants have differing abilities to complement the growth defect of a $\Delta$*secA* mutation

To investigate the effect of the C-terminal truncations on the function of SecA in vivo, we constructed strains in which the sole copy of the *secA* gene produced SecA, SecA$\Delta$MBD or SecA$\Delta$CTT under control of an IPTG-inducible promoter. Because SecA is required for viability, growth of these strains was dependent on the activity of the SecA variant in vivo. All three alleles complemented the viability defect caused by the $\Delta$*secA* mutation (*Figure 2D*) in an IPTG-dependent fashion and produced similar amounts of SecA (*Figure 2—figure supplement 3*), indicating that the truncated proteins were functional in vivo. SecA$\Delta$CTT and SecA$\Delta$MBD were not viable when incubated at room temperature, consistent with the cold-sensitive growth defect of *secB* mutant strains (*Shimizu et al., 1997*). However, cells producing SecA$\Delta$MBD grew poorly even at the permissive temperature (*Figure 2D*), consistent with the idea that truncation of the MBD alone inhibited the activity of SecA.

## Autocrosslinking of the FLD in the substrate binding groove of SecA

In order to affect such a range of activities of SecA, we reasoned that the CTT likely interacts with the catalytic core. To investigate this possibility, we incorporated Bpa into the CTT at positions 852, 893 and 898. In order to distinguish between early termination products and full-length SecA, we fused a short polypeptide tag to the C-terminus of SecA, which causes SecA to be biotinylated in vivo (SecA-biotin) (*Tagwerker et al., 2006*; *Huber et al., 2011*). In addition, we fused hexahistidine-tagged SUMO to the N-terminus of SecA. Ni-affinity purified protein containing Bpa at position 852 (SecA$^{Bpa852}$-biotin) migrated more rapidly than the other proteins in SDS-PAGE (*Figure 4A*), and purified SecA$^{Bpa852}$-biotin interacted with streptavidin indicating it contains the C-terminal biotin (*Figure 4B*). In addition, purification of SecA$^{Bpa852}$-biotin from cell lysates by the C-terminal biotin tag yielded proteins that migrated with molecular weights consistent with full-length SecA-biotin and the faster-migrating species (*Figure 4—figure supplement 1*). These results were consistent with the notion that the faster migrating SecA$^{Bpa852}$-biotin species was the result of an internal crosslink within the protein and not early termination at position 852. The chemical basis for the high efficiency of crosslinking is unknown, but several possible explanations are treated in the Discussion section. Purified SecA$^{Bpa852}$-biotin had a very low affinity for substrate protein and no detectable ATPase activity (*Table 1*), suggesting that SecA$^{Bpa852}$-biotin occupies a conformation similar to that of SecA$\Delta$MBD.

To investigate the site of the internal crosslink, we determined the molecular weights of the tryptic peptides of SecA$^{Bpa852}$-biotin using mass spectrometry (MALDI-TOF). Tryptic fragments with masses greater than 860 Da were resolvable in the mass spectrum of both full-length SecA-biotin and SecA$^{Bpa852}$-biotin (*Supplementary file 1*). Only one peptide in this size range, corresponding to amino acids 168–188, was absent from both spectra. As expected, the tryptic peptide containing position 852 (851-877) was absent from the mass spectrum of SecA$^{Bpa852}$-biotin but not SecA-biotin. The only peptide absent from SecA$^{Bpa852}$-biotin spectrum but present in wild-type SecA-biotin was the peptide corresponding to amino acids 361–382 (*Figure 4C* and *Supplementary file 1*). These results suggested that position 852 likely crosslinked to the region of SecA containing amino acids 361–382. Despite repeated attempts, the crosslinking adduct between peptides 851–877 and 361–382 could not be detected. However, this crosslinking adduct would be very large and would likely consist of a heterogeneous mixture of crosslinked peptides in different conformations. Both of these possibilities could have complicated detection of the adduct by mass spectrometry. Amino acids

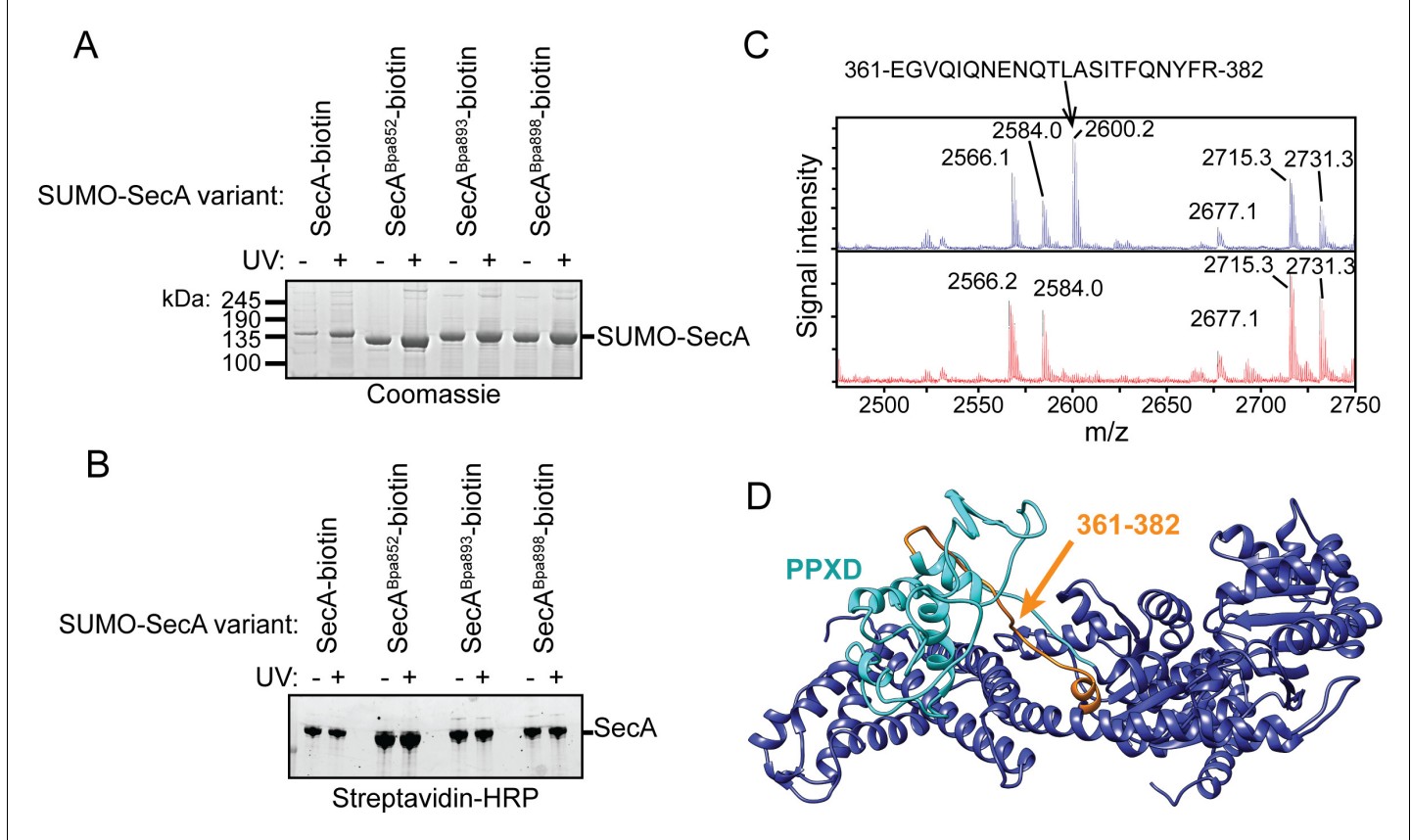

**Figure 4.** Auto-crosslinking of the CTT to the catalytic core of SecA. (**A and B**) 1 µM SUMO-tagged SecA-biotin containing Bpa at position 852, 893 or 898 in the CTT was incubated in the absence (-) or presence (+) of UV light at 365 nm. The protein samples were resolved using SDS-PAGE and visualised by (**A**) Coomassie staining or (**B**) western blotting against the C-terminal biotin tag. The positions of full-length SUMO-SecA is indicated. (**C**) Mass spectra of wild-type SecA-biotin (above, blue) and SecA[Bpa852]-biotin (below, red) in the region of 2450–2750 Da region. Wild-type SecA-biotin and SecA[Bpa852]-biotin were exposed to light at 365 nm and subsequently digested with trypsin. The masses of the tryptic fragments were determined using MALDI-TOF. (**D**) Structure of SecA (2VDA [*Gelis et al., 2007*]). The main body of the catalytic core is coloured blue, the PPXD is coloured cyan and the tryptic peptide that crosslinks to position 852 (amino acids 361–382) is highlighted in orange. The structural model was rendered using Chimera v. 1.12 (*Pettersen et al., 2004*).

DOI: https://doi.org/10.7554/eLife.48385.015

The following figure supplement is available for figure 4:

**Figure supplement 1.** C-terminal purification of SecA-biotin and SecA[Bpa852]-biotin by the C-terminal biotin.
DOI: https://doi.org/10.7554/eLife.48385.016

361–382 are located in one of the two strands linking the PPXD to NBD1 in the groove where SecA binds to substrate protein (*Figure 4D*) (*Cranford-Smith and Huber, 2018*). Crosslinking of Bpa at 852 to this peptide would be consistent with previous work suggesting that the FLD binds in the substrate-binding groove (*Hunt et al., 2002*; *Gelis et al., 2007*).

## Structural analysis of the SecA truncation variants

We next determined the crystal structure of SecAΔMBD at 3.5 Å resolution (6GOX; *Figure 5A* and *Supplementary file 2*). SecAΔMBD crystallised as a symmetric dimer in a head-to-tail configuration (*Figure 5A*). This structure was similar to that reported for the *E. coli* SecA homodimer in complex with ATP (*Papanikolau et al., 2007*) (PDB file 2FSG), except that (i) the PPXD is better resolved in 6GOX and (ii) the 6GOX dimer is symmetric and the 2FSG dimer is not. Consistent with previous studies, the structure of the PPXD was less well defined relative to the other domains of the catalytic core, consistent with the idea that the PPXD is structurally mobile (*Zimmer and Rapoport, 2009*;

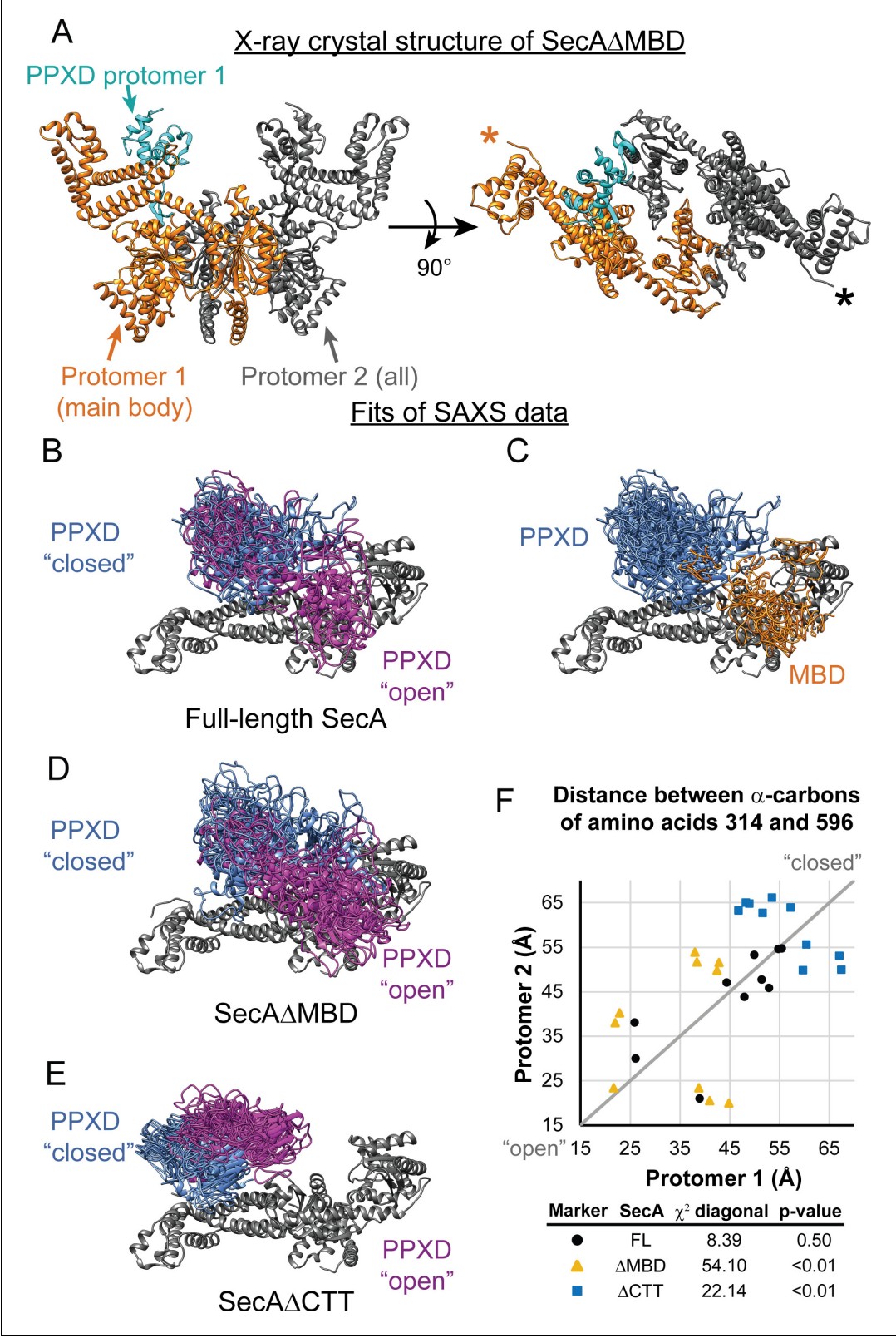

**Figure 5.** SAXS analysis of SecA truncation variants. (**A**) X-ray crystal structure of SecAΔMBD at 3.5 Å solved by molecular replacement. The main body of the catalytic core in the asymmetric unit (Protomer 1) is coloured orange with the PPXD highlighted in cyan. The crystallographic mate (Protomer 2) interacts with promoter one using an interface similar to that found in 2FSG (*Papanikolau et al., 2007*), suggesting that this is the dimer interface of the purified protein in solution. The position of the most C-terminal residue that could be resolved (proline 834) is noted with an asterisk in the right

*Figure 5 continued on next page*

*Figure 5 continued*

panel. (B–E) Overlay of 10 independent structural models of SecA (B, C), SecAΔMBD (D) and SecAΔCTT (E) generated from fitting to the SAXS data using CORAL. The main body of the catalytic core is coloured grey, and the flexible residues are not displayed. (B, D, E) To facilitate visualization of the asymmetry in the in the dimeric models, both protomeric partners of the dimer were overlaid and the PPXD was coloured (blue/magenta) according to the protomer. The MBD is not displayed in panel B. (C) To facilitate visualization of the position of the MBD in the full-length protein, both protomeric partners of the dimer were overlaid and the MBD of the dimer pair that was located nearest to position 596 of the depicted protomer (orange) was displayed. In panel C, the PPXDs of two protomers, which occupy the same space as the MBDs, are not displayed. (F) Plot of the position of the PPXD in partners of the SecA dimer predicted by structural modelling. The distance between the α-carbon of amino acid 314, which is located near the centroid of the PPXD, and amino acid 596 in NBD2 was determined for each protomer and plotted against the distance in the second protomer. SecA, black circles (FL); SecAΔMBD, orange triangles (ΔMBD); SecAΔCTT, blue squares (ΔCTT). The grey diagonal line indicates the position of the distances if the dimers were symmetric. $\chi^2$ values to the diagonal were calculated and used to determine p-values to test whether the asymmetry in the dimer was statistically significant.

DOI: https://doi.org/10.7554/eLife.48385.017

The following figure supplement is available for figure 5:

**Figure supplement 1.** SAXS analysis of the solution structure of SecA, SecAΔMBD and SecAΔCTT.

DOI: https://doi.org/10.7554/eLife.48385.018

*Gold et al., 2013*). However, because the FLD was not resolved, its effect on the structure of SecA could not be determined.

To investigate how the CTT affects the conformation of SecA in solution, we investigated the structures of SecA, SecAΔCTT and SecAΔMBD using small-angle x-ray scattering (SAXS) (*Supplementary file 3*). The SAXS spectra for all three proteins were similar in the low-*q* region, indicating that the overall shapes of the three proteins were similar, and the radii of gyration suggested that they were dimeric, consistent with previous studies (*Woodbury et al., 2002*) (*Figure 5—figure supplement 1*). However, the spectra of the proteins diverged in the mid-*q* region (*Figure 5—figure supplement 1*, arrow), indicating that there were differences in domain organisation. SecA has been crystallised in several distinct dimer configurations. The physiological configuration of the dimer and its relevance is an issue of on-going dispute (see discussion in *Cranford-Smith and Huber, 2018*). However, fitting of structural models of the *E. coli* SecA dimer based on PDB files 2FSG (*Papanikolau et al., 2007*), 2IBM (*Zimmer et al., 2006*), 1M6N (*Hunt et al., 2002*), 1NL3 (*Sharma et al., 2003*), 2IPC (*Vassylyev et al., 2006*) and 6GOX indicated that under the experimental conditions, the conformation of the dimer was similar to that found in 2FSG ($\chi^2$ = 3.66) and 6GOX ($\chi^2$ = 5.25) (*Supplementary file 4*).

To gain insight into the structural differences between the three proteins, we modelled the SAXS data by structural fitting. Because initial fitting runs indicated that the CTT was in close proximity to the catalytic core in models of full-length SecA and SecAΔMBD, we fixed the position of the FLD in subsequent fits so that it was consistent with the Bpa crosslinking results. The resulting models suggested that the PPXD was positioned considerably closer to NBD2 (i.e. more 'open') in SecA and SecAΔMBD than in SecAΔCTT (p=2.0×10$^{-5}$ and 1.1 × 10$^{-7}$, respectively) (*Figure 5F*). In models of SecAΔMBD and SecAΔCTT, the PPXDs in the two protomers of the dimers were positioned asymmetrically (p=1.8×10$^{-8}$ and 0.0085, respectively) (*Figure 5B–D,F*). Finally, in models of full-length SecA, the MBD was positioned between NBD2 and the C-terminal portion of the HSD (amino acids 756–832) in both protomers of the dimer (*Figure 5E*). Localisation of the MBD to this region would position it directly adjacent to the ribosome-binding surface on the catalytic core (*Huber et al., 2011*; *Singh et al., 2014*).

## Discussion

Our results indicate that the CTT controls the conformation of SecA and regulates its activity. Disruption of the MBD alone (i) increases the affinity of SecA for the ribosome, (ii) decreases the affinity of SecA for substrate protein, (iii) inhibits the ATPase activity of SecA, (iv) increases the thermal stability of SecA, (v) prevents SecA from undergoing a conformational change upon binding to nascent substrate protein and (vi) causes a defect in SecA function in vivo. However, disruption of both the MBD and the FLD results in a protein that behaves very similarly to full-length SecA, indicating that the FLD mediates these effects. Chemical crosslinking and structural modelling of the SAXS data for

wild-type SecA suggest that the FLD interacts with the catalytic core (potentially binding in the substrate protein binding groove) and causes a conformational change in the PPXD. *Gold et al. (2013)* have suggested that opening of the PPXD when SecA is bound to substrate protein (*i.e.* enclosing the substrate protein within the binding groove by the PPXD clamp) activates the ATPase activity of SecA. Our work suggests that enclosure of the FLD by the PPXD has the opposite effect—that is, autoinhibition of SecA.

We propose that the MBD is the key for unlocking this autoinhibited conformation in the full-length protein. Previous work suggests that interaction of the MBD with SecB increases the affinity of SecA for polypeptides (*Gelis et al., 2007*). Our results raise the possibility that binding of the MBD to the ribosome activates SecA in a similar fashion. The absence of the MBD does not cause a strong defect in binding of SecA to ribosomes (indeed, SecAΔMBD has a higher affinity for ribosomes than full-length SecA and SecAΔCTT), and our results suggest that binding of the catalytic core to the ribosome would place the MBD in an ideal position to bind to the ribosomal surface. Thus, although the affinity of the MBD in isolation for the ribosome is relatively low, binding of the catalytic core could trigger binding of the MBD to the ribosome in the context of the full-length protein.

Taken together, our results allow us to propose a mechanism for the recognition of nascent substrate proteins by SecA (*Figure 6*): (i) interaction of the MBD with the ribosomal surface upon binding of the catalytic core of SecA to the ribosome destabilises the interaction between the FLD and the catalytic core; (ii) destabilisation of the FLD allows SecA to sample nascent polypeptides; (iii) the stable interaction of SecA with nascent substrate protein displaces the FLD from the substrate binding groove; and (iv) binding of SecA to nascent substrate protein causes a conformational change in SecA that leads to release from the ribosome.

The physiological role of CTT-mediated autoinhibition is not yet known. One possibility is that autoinhibition prevents the spurious interaction of SecA with non-substrate proteins by only allowing it to interact with polypeptides in the presence of its ligands (*e.g.* translating ribosomes, SecB and

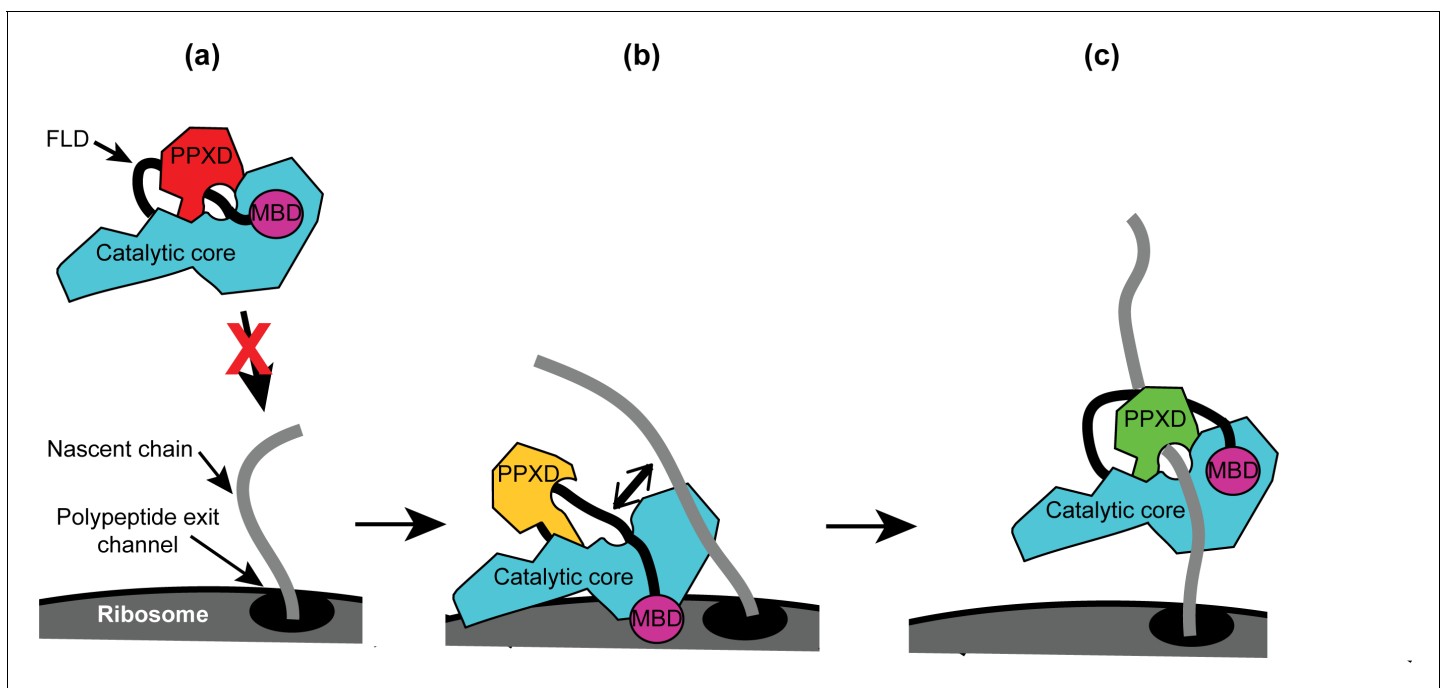

**Figure 6.** Diagram of the proposed mechanism for recognition of nascent substrate proteins by SecA. (a) In solution, SecA occupies an autoinhibited conformation with the FLD bound stably in the substrate protein binding site and the PPXD in the open conformation. (b) Binding of both the catalytic core and the MBD to the ribosomal surface causes the PPXD to shift to the open conformation, which destabilises binding of the FLD and allows SecA to sample nascent polypeptides. (c) Binding to the nascent substrate protein displaces the FLD from the substrate protein binding site and the PPXD returns to the open conformation, stabilising this interaction. Binding to nascent substrate releases SecA from the ribosomal surface.
DOI: https://doi.org/10.7554/eLife.48385.019

potentially phospholipids/SecYEG) (*Breukink et al., 1995*; *Gelis et al., 2007*; *Fekkes et al., 1999*). Indeed, overproduction of substrate polypeptides causes a translocation defect in vivo (*Wagner et al., 2007*; *Müller et al., 1989*; *Oliver and Beckwith, 1982*), indicating that SecA can be overwhelmed by interactions with too many substrate proteins. In addition, the spurious translocation of cytoplasmic proteins can be toxic (*van Stelten et al., 2009*; *Emr et al., 1978*). Alternatively, the structure of the MBD could regulate the activity of SecA in response to physiological stress. For example, research by the Huber group suggests that the physiological ligand of the MBD is iron (BIORXIV/2019/613315). It is possible that the structure of the MBD is regulated in response to iron limitation or the redox state of the bound metal. If so, the partial activity of SecA∆MBD in vivo suggests that the CTT modulates the activity of SecA rather than inhibiting it completely.

Our results suggest that SecA[Bpa852]-biotin produces auto-crosslinks very efficiently. At least three factors could contribute to the high efficiency of auto-crosslinking in SecA[Bpa852]-biotin: (i) the amount of time the benzophenone group of the Bpa is in contact with the target molecule, (ii) the chemical reactivity of Bpa toward the target molecule, and (iii) the amount of time the benzophenone group stays in the activated state. First, the results of this study and others (*Gelis et al., 2007*) is consistent with the idea that the FLD is stably bound in the substrate protein-binding groove of SecA, which should result in a long-lived contact between position 852 and the amino acids lining the substrate binding groove. Second, although benzophenone can, in theory, react with any C-H bond, in practice it reacts with different efficiencies toward different amino acid side chains (*Wittelsberger et al., 2006*; *Lancia et al., 2014*). Finally, the chemical environment surrounding a benzophenone group (*e.g.* hydrophobicity, pH, *etc.*) can influence its photo-reactive properties (*Barsotti et al., 2015*; *Barsotti et al., 2017*). Because Bpa is typically incorporated at surface-exposed positions in order to capture protein-ligand interactions, these environmental effects are normally negligible. However, the hydrophobic environment surrounding the side chain of position 852 when bound in the substrate binding groove could have a significant effect on its reactivity.

The basic features of the catalytic core of SecA are highly conserved amongst bacteria, but different bacterial species contain a diverse array of loops and extensions. For example, our phylogenetic analysis indicated that many species contain alternative CTTs with structures that are significantly different from that of *E. coli*. These differences could allow SecA to be regulated in response to interaction with a different subset of interaction partners. Nonetheless, many of these alternative CTTs are highly positively charged (*e.g.* those of many Actinobacteria), suggesting that they may retain the interaction with the ribosome. Some phylogenetic groups (*e.g.* the Cyanobacteria) lack a CTT entirely. However, most of these species contain large loops in between the conserved elements of the catalytic core of the protein. Indeed, *E. coli* SecA also contains a 'variable' subdomain in NBD2 (amino acids ~ 519–547), which has been proposed to regulate its activity (*Das et al., 2012*). It is possible that the large loops in between the conserved features of the catalytic core could function analogously to the CTT in *E. coli* SecA.

## Materials and methods

### Key resources table

| Reagent type (species) or resource | Designation | Source or reference | Identifiers | Additional information |
|---|---|---|---|---|
| Strain, strain background (*Escherichia coli K-12*) | MC4100 | *Casadaban, 1976* | | F⁻ *araD139* D*lacU169* *rpsL150 thi rbsR* |
| Strain, strain background (*Escherichia coli K-12*) | DRH1119 | This paper | | MC4100 ∆*secA*:: Kan^R λ_att::p_lacUV5-*secA* |
| Strain, strain background (*Escherichia coli K-12*) | DRH1120 | This paper | | MC4100 ∆*secA*:: Kan^R λ_att::p_lacUV5⁻ *secA∆MBD* |
| Strain, strain background (*Escherichia coli K-12*) | DRH1121 | This paper | | MC4100 ∆*secA::Kan^R* λ_att::p_lacUV5-*secA∆CTT* |

*Continued on next page*

*Continued*

| Reagent type (species) or resource | Designation | Source or reference | Identifiers | Additional information |
|---|---|---|---|---|
| Strain, strain background (*Escherichia coli K-12*) | DRH663 | This paper | | MC4100 ΔsecA ::Kan$^R$ + pDH663 |
| Strain, strain background (*Escherichia coli B*) | BL21(DE3) | Lab stock | *Escherichia coli* Genetic Stock Center (CGSC), Yale University, USA. CGSC#: 12504 http://cgsc2.biology .yale.edu/Strain.php? ID=139459 | MC4100 ΔsecA ::Kan$^R$ + pDH663 |
| Strain, strain background (*Escherichia coli B*) | DRH584 | This paper | | BL21(DE3) containing plasmid pDH584 |
| Strain, strain background (*Escherichia coli B*) | DRH1166 | This paper | | BL21(DE3) containing plasmid pDH1166 and pSup-Bpa-6TRN |
| Genetic reagent (phage λ) | lambda InCh | *Boyd et al., 2000* | | |
| Antibody | Rabbit anti-SecA antiserum | *Huber et al., 2011* | | (1:20000) |
| Antibody | Sheep anti-uL23 antiserum | other | | Gift from R. Brimacombe (1:2500) |
| Antibody | Rabbit anti-thioredoxin-1 antiserum | Sigma-Aldrich (St. Louis, MO, USA) | Catalogue number: T0803 | (1:10000) |
| Antibody | Goat IR700-labelled anti-rabbit | Rockland (Philadelphia, PA, USA) | Catalogue number: 611-130-122 | Discontinued (1:5000) |
| Antibody | Goat IR800-labelled anti-sheep | Rockland (Philadelphia, PA, USA) | Catalogue number: 613-445-002 | (1:10000) |
| Antibody | Donkey HRP-labelled anti-rabbit | GE Healthcare | Catalogue number: NA934V | (1:10000) |
| Recombinant DNA reagent | pCA528 | *Andréasson et al., 2010* | | pET24 expression vector containing gene encoding His-tagged SUMO protein |
| Recombinant DNA reagent | pCA597 | *Andréasson et al., 2010* | | pET24 expression vector containing gene encoding Strep-tagged SUMO protein |
| Recombinant DNA reagent | pDH543 | This paper | | pCA597 containing portion of *secA* gene corresponding to amino acids 829–901 |
| Recombinant DNA reagent | pDH934 | This paper | | pCA597 containing portion of *secA* gene corresponding to amino acids 875–901 |
| Recombinant DNA reagent | pDH625 | *Huber et al., 2011* | | pCA528 containg the *secA* gene |

*Continued on next page*

*Continued*

| Reagent type (species) or resource | Designation | Source or reference | Identifiers | Additional information |
|---|---|---|---|---|
| Recombinant DNA reagent | pDH584 | *Huber et al., 2011* | | pCA528 containg the *secA-biotin* gene |
| Recombinant DNA reagent | pDH1166 | This paper | | pDH584 containing amber codon at position corresponding to amino acid 852 in SecA. |
| Recombinant DNA reagent | pDSW204 | *Weiss et al., 1999* | | pTrc99a-derived plasmid containing partially disabled *trc* promoter |
| Recombinant DNA reagent | pDH692 | *Huber et al., 2011* | | pDSW204-derived plasmid producing SecA under control of an IPTG-inducible promoter |
| Recombinant DNA reagent | pDH939 | This paper | | pDSW204-derived plasmid producing SecAΔMBD under control of an IPTG-inducible promoter |
| Recombinant DNA reagent | pDH663 | *Huber et al., 2011* | | pTrc99b-derived plasmid producing SecA under control of an IPTG-inducible promoter and containing a Spectinomycin resistance gene in place of the *bla* (ampicillin resistance) gene |
| Recombinant DNA reagent | pDH787 | *Huber et al., 2017* | | pCA597 containing full-length *secM* gene |
| Recombinant DNA reagent | pDH784 | *Huber et al., 2017* | | pCA597 containing *secM* with internal deletion between region encoding signal sequence and the translation arrest sequence |
| Recombinant DNA reagent | pDH894 | *Huber et al., 2017* | | pCA528 containing gene encoding *malE* gene that is translationally fused to sequence encoding the SecM translation arrest sequence |
| Recombinant DNA reagent | pSup-Bpa-6TRN | *Chin et al., 2002* | | Plasmid producing orthologous tRNA and tRNA synthetase required for incorporation of Bpa |

*Continued on next page*

*Continued*

| Reagent type (species) or resource | Designation | Source or reference | Identifiers | Additional information |
|---|---|---|---|---|
| Peptide, recombinant protein | VipB peptide | *Pietrosiuk et al., 2011* | | |
| Peptide, recombinant protein | HRP-coupled Streptactin | IBA Life Sciences (Goettingen, Germany) | Catalogue number: 2-1502-001 | |
| Chemical compound | 4-acetamido-4'-maleimidylstilbene-2, 2'-disulfonic acid (AMS) | Invitrogen (Carlsbad, California) | Catalogue number: A485 | |
| Chemical compound, drug | Hydrophilic streptavidin magnetic beads | New England Biolabs (Ipswich, Massachusetts) | Catalogue number: S1421S | |
| Chemical compound, drug | Benzophenylalanine (Bpa) | Bachem (Santa Cruz, CA, USA) | H-p-Bz-Phe-OH Article number: 4017646.0005 | |
| Chemical compound, drug | 1-ethyl-3-(3-dimethylaminopropyl) carbodiimide (EDC) | ThermoScientific Pierce | Catalogue number: 22980 | |
| Chemical compound, drug | Ru(bpy)2(dcbpy) | Sigma Aldrich (St Louis, MO, USA) | Product 96632 | Discontinued |
| Software, algorithm | ATSAS v2.8.3 | European Molecular Biology Laboratory (EMBL) Hamburg | https://www.embl-hamburg.de/biosaxs/software.html | |
| Software, algorithm | PyMol v1.8.0.5 | Schrödinger Scientific | https://pymol.org/2/ | |
| Software, algorithm | GROMACS | Schrödinger Scientific | *Pronk et al., 2013* | |
| Chemical compound, drug | PEG/Ion Screen 2 #39 | Hampton Research (Aliso Viejo, CA, USA) | Product HR2-126 | |
| Chemical compound, drug | Morpheus | Molecular Dimensions (Newmarket, Suffolk, UK) | Product MD1-46 | |
| Other | Superose 6 10/300 GL column | GE Healthcare | Product 17517201 | Discontinued |

## Chemicals and media

All chemicals were purchased from Sigma-Aldrich (St. Louis, MO, USA) unless otherwise indicated. Anti-thioredoxin-1 antiserum was purchased from Sigma-Aldrich. Rabbit anti-SecA antiserum was a laboratory stock. Sheep anti-uL23 antiserum was a kind gift from R. Brimacombe. IR700-labelled anti-rabbit and IR800-labelled anti-sheep antibodies were purchased from Rockland (Philadelphia, PA). Horseradish peroxidase (HRP)-labelled anti-rabbit antibody was purchased from GE Healthcare. HRP-coupled streptactin was purchased from IBA Lifesciences (Goettingen, Germany). Bpa was purchased from Bachem (Santa Cruz, CA). Strains were grown in lysogeny broth (LB) containing kanamycin (30 µg/ml) or ampicillin (200 µg/ml) as required and in the concentration of isopropyl-thiogalactoside (IPTG) indicated.

## Strains and plasmids

Strains and plasmids were constructed using standard methods (*Miller, 1992*; *Sambrook and Russell, 2001*). For protein-expression plasmids, the DNA encoding full-length SecA or fragments of SecA were amplified by PCR and ligated into plasmid pCA528 (His$_6$-SUMO) or pCA597 (Strep$_3$-SUMO) using the *Bsa*I and *Bam*HI restriction sites (*Andréasson et al., 2008*). UAG stop codons were introduced into plasmids expressing His-SUMO-SecA or His-SUMO-SecA-biotin using QuikChange

(Agilent). Plasmid pSup-Bpa-6TRN was a kind gift from P Schultz. Strains DRH1119, DRH1120 and DRH1121 were constructed by cloning *secA* genes producing full-length SecA, SecAΔMBD and SecAΔCTT into pDSW204 (*Weiss et al., 1999*) and then introducing them onto the chromosome strain DRH663 (MC4100 Δ*secA*::Kan$^R$ + pTrcSpc-SecA) (*Huber et al., 2011*) using lambda InCh (*Boyd et al., 2000*). The pTrcSpc-SecA plasmid was then cured from the strain by plating on LB containing 1 mM IPTG. All three strains required >10 μM IPTG for growth on LB.

## Phylogenetic analysis

The sequences of SecA for the given UniProtKB entry names (*The UniProt Consortium, 2017*) were analysed using ClustalOmega (*McWilliam et al., 2013*). The unrooted phylogenetic tree was rendered using iTOL (*Ciccarelli et al., 2006*). The logo of the consensus MBD sequence was generated using WebLogo (https://weblogo.berkeley.edu/logo.cgi).

## Ribosome and protein purification

Ribosomes and arrested RNCs were purified as previously described (*Rutkowska et al., 2008*; *Huber et al., 2011*). SecA was purified as described previously (*Huber et al., 2017*). BL21(DE3) (laboratory stock) or BL21(DE3) containing plasmid pSup-Bpa-6TRN (*Chin et al., 2002*) was transformed with the appropriate plasmid and grown in LB in the presence of kanamycin at 37°C to OD$_{600}$ ~1, induced using 1 mM IPTG and shifted to 18°C overnight. Cells were then harvested by centrifugation and lysed by cell disruption in buffer 1 (50 mM K·HEPES, pH 7.5, 500 mM NaCl and 0.5 mM TCEP [tris(2-carboxyethyl)phosphine]) containing cOmplete EDTA-free protease inhibitor cocktail (Roche). Unlabelled His-tagged proteins were affinity purified by passing over a 5 ml Ni-NTA HiTrap column (GE Healthcare), washed with buffer containing 50 mM imidazole and eluted from the column in buffer containing 250 mM imidazole. The eluted protein was cleaved with the SUMO-protease Ulp1 and the SUMO moiety was removed by passing over a 5 ml Ni-NTA HiTrap column. The partially purified protein was then concentrated (Centricon) and purified by size exclusion chromatography using a sepharose S-200 column (GE Healthcare). Bpa-labelled proteins were purified as described by *Huber et al. (2017)*. For Strep-tagged proteins, lysates from cells producing SUMO-CTT and SUMO-MBD were passed over streptactin-coupled sepharose beads (IBA Lifesciences), washed extensively with buffer 2 (20 mM K·HEPES, pH 7.5, 100 mM potassium acetate, 10 mM magnesium acetate) and eluted using buffer 2 containing 10 mM desthiobiotin. SUMO-CTT was modified with 4-acetamido-4′-maleimidylstilbene-2,2′-disulfonic acid (AMS) by incubating 75 μM SUMO-CTT with 500 μM AMS in buffer 2 for 30 min on ice. AMS labelling was terminated by the addition of 500 μM β-mercaptoethanol. Efficient labelling was confirmed by the increased mobility of the modified protein in SDS-PAGE. For purification of SUMO-SecA$^{Bpa852}$-biotin by its C-terminal biotin tag, lysates of cells producing SUMO-SecA-biotin and SUMO-SecA$^{Bpa852}$-biotin were incubated with 50 μl hydrophilic streptavidin magnetic beads and washed five times with 1 ml buffer 2. The bound protein was eluted from the beads by boiling in 50 μl 1X Laemmli buffer and resolved on a 12% BioRad Stain-free Ready gel.

## Western blotting

Western blots were carried out as previously described (*Sambrook and Russell, 2001*). Protein samples were resolved using 'Any kD' SDS-PAGE gels (BioRad) and transferred to nitrocellulose membranes. Membranes were probed using the indicated primary and secondary antisera or with HRP-streptactin. For HRP-based detection, membranes were developed using ECL (GE Healthcare) and visualised using a BioRad Gel-Doc. For IR700- and IR800-based detection, membranes were visualised using a LI-COR Odyssey scanner.

## Ribosome cosedimentation

Ribosome cosedimentation experiments were carried out as previously described (*Huber et al., 2017*). Binding reactions were incubated in buffer containing 10 mM HEPES potassium salt, pH 7.5, 100 mM potassium acetate, 10 mM magnesium acetate, 1 mM β-mercaptoethanol for >10 min. The reaction mixture was then layered on top of a 30% sucrose cushion made with the same buffer and centrifuged at >200,000 x *g* for 90 min. The supernatant was discarded. The concentration of ribosomes in the pellet fractions were normalised using the absorbance at 260 nm.

## Fluorescence anisotropy

The $K_D$ of the SecA-ribosome complex by fluorescence anisotropy was determined as previously described (*Huber et al., 2011*). SecA, SecAΔMBD and SecAΔCTT were labelled with Ru (bpy)$_2$(dcbpy) and the fluorescence anisotropy was measured on a Jasco FP-6500 fluorometer containing an ADP303 attachment using an excitation wavelength of 426 nm (slit width 5 nm) and an emission wavelength of 640 nm (slit width 10 nm).

## CD spectroscopy

The CD spectra of 2 µM solutions of full-length SecA, SecAΔMBD, or SecAΔCTT in 10 mM potassium phosphate buffer (pH 7.5) were measured at temperatures that promote folding (10˚C) and denaturation (85˚C) in a 0.1 cm cuvette using a Jasco J750 CD spectrometer. For thermal titrations, the temperature was raised 0.5 K/min from 30˚C to 50˚C and circular dichroism was measured at 222 nm.

## Peptide binding

600 nM VipB peptide labelled with IAANS (*Pietrosiuk et al., 2011*) was incubated with increasing concentrations of SecA or the indicated SecA variant. The increase in IAANS fluorescence upon binding of SecA was measured using a Jasco FP-6500 fluorometer or a BMG Labtech CLARIOStar.

## ATPase assays

ATPase activities were determined by measuring the rate of NADH oxidation in a coupled reaction (*Kiianitsa et al., 2003*). 1 µM SecA, or the respective SecA variant, was added to a solution containing 250 mM NADH, 0.5 mM phosphoenolpyruvate, 2 mM ATP, 20/ml lactate dehydrogenase, 100 U/ml pyruvate kinase and incubated at 25˚C, 50 mM K·HEPES, pH 7.5 and 500 mM NaCl. The decrease in absorbance at 340 nm from the oxidation of NADH to $NAD^+$ was measured using an Anthos Zenyth 340rt (Biochrom) absorbance photometer equipped with ADAP software. The rate of ATP hydrolysis was determined from the rate of NADH oxidation by dividing the rate of decrease in the absorbance by the extinction coefficient for NADH (6220 $M^{-1}$ $cm^{-1}$).

## Chemical crosslinking

Non-specific crosslinking of SUMO-CTT to the ribosome using 1-ethyl-3-(3-dimethylaminopropyl) carbodiimide (EDC) were carried out as described previously (*Huber et al., 2011*). Site specific crosslinking of SecA containing Bpa was carried out as described by *Huber et al. (2017)*.

## Mass spectrometry

Auto-crosslinked samples were digested with sequencing grade trypsin and the masses of the tryptic peptides were determined using MALDI-TOF mass spectrometry. The identity of ribosomal crosslinking adducts was determined by excising the protein band from a Coomassie-stained gel and analysing the tryptic peptide fragments using liquid chromatography-tandem mass spectrometry (LC-MS/MS) identification (The Advanced Mass Spectrometry Facility, School of Biosciences, University of Birmingham).

## X-ray crystallography

SecAΔMBD was crystallised by mixing 2 µl of purified protein (100 µM) with 2 µl of a 9:1 mixture of PEG/Ion 2 Screen # 39 buffer (0.04M Citric acid, 0.06M BIS-TRIS propane, pH6.4, 20% Polyethylene glycol 3,350) (Hampton Research, Aliso Viejo, CA, USA) and Morpheus condition 47, box 2 (0.1M Amino acids, 0.1M Buffer system 3, pH 8.5, 50% v/v Precipitant Mix 3) (Molecular Dimensions, Newmarket, UK) in a 48-well MRC MAXI plate (Molecular Dimensions, Newmarket, UK). Crystals appeared within six days and were fully matured by two weeks. Crystals were analysed at Diamond light source, and the structure was solved by molecular replacement using PDB file 2FSG at 3.5 Å resolution (*Figure 1—source data 1*). The structure was deposited at RCSB under PDB file 6GOX.

## SAXS measurements

Synchrotron radiation X-ray scattering data were collected on the ESRF BM29 BioSAXS beamline (Grenoble) (*Figure 1—source data 2*). An in-line Superose 6 10/300 GL column (GE Healthcare) was used to ensure that the protein was free from aggregates and that it occupied a single oligomeric

state during data collection. The sample-to-detector distance was 3 m, covering a range of momentum transfer s = 0.03–0.494 $\text{Å}^{-1}$ (s = (4π·sin θ)/ λ, where 2θ is the scattering angle, and λ = 0.992 Å is the X-ray wavelength). Data from the detector were normalised to the transmitted beam intensity, averaged, placed on absolute scale relative to water and the scattering of buffer solutions subtracted. All data manipulations were performed using PRIMUS*qt* and ATSAS (*Petoukhov et al., 2012*). The forward scattering *I(0)* and radius of gyration, $R_g$ were determined by Guinier analysis. These parameters were also estimated from the full scattering curves using the indirect Fourier transform method implemented in the program GNOM, along with the distance distribution function *p(r)* and the maximum particle dimensions $D_{max}$. Molecular masses of solutes were estimated from SAXS data by comparing the extrapolated forward scattering with that of a reference solution of bovine serum albumin. Computation of theoretical scattering intensities was performed using the program CRYSOL. SAXS data has been deposited at the SASBDB (www.sasbdb.org) with accession codes: SASDDY9 (full-length SecA), SASDDZ9 (SecAΔMBD) and SASDE22 (SecAΔCTT).

## Molecular modelling of SAXS data

For modelling based on SAXS data, multiple fits were performed to verify the stability of the solution, and to establish the most typical 3D reconstructions using DAMAVER. Guinier analysis of the SAXS data indicated that the protein was dimeric under the conditions used for SAXS. Structural models of the *E. coli* SecA dimer were generated by aligning the structure of SecA in PDB file 2VDA to PDB files 2FSG, 2IBM, 2IPC, 1M6N, 1NL3 and 6GOX using PyMol v. 1.8.0.5 and refining using GROMACS (*Pronk et al., 2013*). Because the CTT is not resolved in the structures used for modelling, these models were fitted to the SAXS spectrum of SecAΔCTT using FoXS (*Schneidman-Duhovny et al., 2016*) (*Supplementary file 1*). The structures of full-length SecA, SecAΔMBD and SecAΔCTT were modelled by fitting the 2FSG dimer to the respective SAXS data by multi-step rigid body refinement using CORAL (*Petoukhov et al., 2012*). The positions of NBD1, NBD2, HSD and HWD were fixed in all models. The regions corresponding to residues 1–8, 220–231, 367–375 were defined as linkers and modelled as flexible. The PPXD was allowed rigid body movement in all three models. The FLD (residues 829–832 in SecAΔCTT and 829–880 in SecAΔMBD and full-length SecA) were modelled as flexible. For full length SecA, the MBD was modelled using PDB file 1S × 0 and allowed rigid-body movement. Because initial modelling indicated that the FLD was in close contact with the catalytic core, and because photocrosslinking indicated the FLD was bound in the substrate binding groove, residues 851–854 were modelled to form a small β-sheet with residues 222–225 and 373–375 and allowed rigid body movement. All ten independently generated fits for SecA and SecAΔMBD produced plausible structural models ($\chi^2$ = 0.97 ± 0.02 and 1.87 ± 0.09, respectively). Ten of 15 of the fits of SecAΔCTT produced plausible structural models ($\chi^2$ = 1.82 ± 0.18). In the remaining five fits, the position of the PPXD in one of the two protomers was inconsistent with previously published structures of SecA and occupied a non-realisitic conformation, suggesting increased mobility of the PPXD in SecAΔCTT.

## Acknowledgements

We thank A McNally and members of the Bukau, Mayer, Henderson, Lund, Grainger, Cole and Rossiter groups for helpful advice and discussions. We thank B Zachmann-Brand, Jingli Yu and the functional genomic facility (School of Biosciences, University of Birmingham) for technical assistance. We are grateful to the University of Birmingham Protein Expression Facility for use of their facilities.

## Additional information

### Funding

| Funder | Grant reference number | Author |
|---|---|---|
| Biotechnology and Biological Sciences Research Council | BB/L019434/1 | Mohammed Jamshad Damon Huber |
| Biotechnology and Biological Sciences Research Council | BB/P009840/1 | Timothy J Knowles Gareth W Hughes |

| Biotechnology and Biological Sciences Research Council | MIBTP | Max Wynne |
| --- | --- | --- |
| Deutsche Forschungsgemeinschaft | FOR 1805 | Günter Kramer Bernd Bukau |
| Deutsche Forschungsgemeinschaft | SFB 638 | Günter Kramer Bernd Bukau |
| Wellcome | 099266/Z/12/Z | Fiyaz Mohammed |
| Deutsche Forschungsgemeinschaft | KR3593/2-1 | Günter Kramer |

The funders had no role in study design, data collection and interpretation, or the decision to submit the work for publication.

## Author contributions

Mohammed Jamshad, Supervision, Investigation, Methodology, Writing—review and editing; Timothy J Knowles, Resources, Data curation, Validation, Investigation, Methodology; Scott A White, Data curation, Formal analysis, Investigation, Writing—review and editing; Douglas G Ward, Max Wynne, Investigation, Writing—review and editing; Fiyaz Mohammed, Resources, Investigation, Writing—review and editing; Kazi Fahmida Rahman, Investigation, Methodology, Writing—review and editing; Gareth W Hughes, Investigation; Günter Kramer, Conceptualization, Resources, Funding acquisition, Writing—review and editing; Bernd Bukau, Funding acquisition, Writing—review and editing; Damon Huber, Conceptualization, Supervision, Funding acquisition, Investigation, Methodology, Writing—original draft, Project administration, Writing—review and editing

## Author ORCIDs

Gareth W Hughes (iD) http://orcid.org/0000-0002-1228-6152
Günter Kramer (iD) http://orcid.org/0000-0001-7552-8393
Bernd Bukau (iD) http://orcid.org/0000-0003-0521-7199
Damon Huber (iD) https://orcid.org/0000-0002-7367-3244

## Decision letter and Author response

Decision letter https://doi.org/10.7554/eLife.48385.034
Author response https://doi.org/10.7554/eLife.48385.035

# Additional files

## Supplementary files

• Supplementary file 1. Table of SecA tryptic peptides detected by MALDI-TOF.
DOI: https://doi.org/10.7554/eLife.48385.020

• Supplementary file 2. Table of data collection and refinement statistics for the crystal structure of SecAΔMBD.
DOI: https://doi.org/10.7554/eLife.48385.021

• Supplementary file 3. Table of SAXS data collection and processing details for SecA, SecAΔMBD and SecAΔCTT.
DOI: https://doi.org/10.7554/eLife.48385.022

• Supplementary file 4. Table of fitting parameters of models of the *E. coli* SecA dimer.
DOI: https://doi.org/10.7554/eLife.48385.023

• Transparent reporting form
DOI: https://doi.org/10.7554/eLife.48385.024

## Data availability

X-ray crystallography data are deposited in PDB under accession code 6GOX. Small-angle x-ray scattering data are deposited in SASBDB under accession codes SASDDY9, SASDDZ9 and SASDE22.

The following datasets were generated:

| Author(s) | Year | Dataset title | Dataset URL | Database and Identifier |
| --- | --- | --- | --- | --- |
| Huber D, White S, Jamshad M | 2018 | SecA | http://www.rcsb.org/structure/6GOX | Protein Data Bank, 6GOX |
| Knowles T, Jamshad M, Huber D | 2018 | SecA | https://www.sasbdb.org/data/SASDDY9 | Small Angle Scattering Biological Data Bank, SASDDY9 |
| Knowles T, Jamshad M, Huber D | 2018 | SecAΔMBD | https://www.sasbdb.org/data/SASDDZ9 | Small Angle Scattering Biological Data Bank, SASDDZ9 |
| Knowles T, Jamshad M, Huber D | 2018 | SecAΔCTT | https://www.sasbdb.org/data/SASDE22 | Small Angle Scattering Biological Data Bank, SASDE22 |

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
