## [Decision Letter]

[Editors’ note: a previous version of this study was rejected after the first round of revisions, but the authors submitted for reconsideration. The first decision letter after peer review is shown below.]

Thank you for submitting your article "The C-terminal tail of the bacterial translocation ATPase SecA modulates its activity" for consideration by *eLife*. Your article has been reviewed by three peer reviewers, and the evaluation has been overseen by a Reviewing Editor and John Kuriyan as the Senior Editor. The reviewers have opted to remain anonymous.

The reviewers have discussed the reviews with one another and the Reviewing Editor has drafted this decision to help you prepare a revised submission.

There are two main issues to be addressed experimentally:

1) Experiments to determine whether and which divalent metal(s) impact function of the C-terminal domain. A complete analysis is probably beyond the scope of this paper, but initial experiments to examine the metal requirement were judged to be important.

2) Disentangling the role of the nascent chain versus a programmed ribosome by analysis of RNCs containing short nascent chains.

In addition, a number of clarifications, controls, and fuller explanations were requested (see the full reviews appended below). Each of these seem relatively straightforward. I suspect that for many of these, you will already have the answer from experiments not shown in the manuscript, although some might also require repeating experiments with some variations.

Finally, it is worth addressing the issue of the interplay between the SecA-nascent chain interaction studied here as it relates to known interactions of SecA with lipids and SecYEG. Experimental investigation of this issue is beyond the scope of this study, but discussing this issue is worthwhile.

*Reviewer #1:*

The manuscript submitted by the Huber lab relates to the regulation of the essential motor ATPase responsible for the bulk of bacterial protein secretion. The results build up to a hitherto unknown mechanism for its auto-inhibition and activation by nascent substrates at the ribosome exit site. The submission compiles high quality data, which have been sensibly interpreted, so the story is plausible. It is a story that should certainly be worth considering for those of us interested in SecA, and who have until now completely ignored the regulatory feature of the C-terminus. The paper will also be of interest to a much wider readership, and also for folk interested in the development of antibiotics. Thus, the work could be worthy of publication in *eLife* if the following points and suggestions can be addressed/considered.

To be addressed:

1) The metal binding site seems to have an important role here. Does it require metals to function? If so, which ones – Zn^2+^? Probably some/many of the experiments should be compared with and without divalent metals.

2) Figure 3A shows important evidence for interaction of flexible linker with the catalytic core by intra-molecular crosslinking of residue 852. The authors say the SecA-852 band is fully crosslinked, in contrast to 893 and 898, because of its higher apparent mobility. I would like the authors to redo this experiment, but with equal quantities of SecA, just to check that it is not a loading artefact. Also, why is the crosslink not induced by UV? Were the uncrosslinked samples kept in the dark at all times?

3) Concerning the X-ray structure. If the FLD influences the conformation of SecA, as stated, then it should be evident in the high-resolution structure. How does the new structure compare to others determined without the FLD? Did the full-length protein fail to crystallise? This figure could do with some improvement by colouring the different domains and highlighting the location of the C-terminus and FLD. The two monomers of the dimers could also be distinguished somehow.

*Reviewer #2:*

The study from Jamshad and colleagues investigates the function of the C-terminal domain of SecA, a key factor in protein export in bacteria. The authors show that in *E. coli* the C-terminal domain of SecA, as well as binding SecB, can also bind to the ribosome. Loss of the entire CTT leads to no obvious growth defect and actually enhances ribosome binding, suggesting that it likely has a regulatory function. Indeed through crosslinking, structural and functional analyses a model emerges where the metal binding domain of the CTT acts to control substrate loading in response to ribosome binding. Overall, the experiments are largely well performed and the study provides new insight into SecA function. Following attention to the points below I would be supportive of publication.

In Figure 1D the SUMU-CTT is in large excess over ribosomes, it would be important to see a 10% input to assess the efficiency of binding. It would also be informative to assess whether SecA lacking the CTT is able to inhibit binding.

In Figure 2B, the effect of the nascent chain is established by comparing empty 70S ribosomes with a stalled SecM RNC. This could reflect changes due to the nascent chain, but also the fact that the ribosomes are now programmed. This could be distinguished using a shorter nascent chain that has not yet emerged from the exit tunnel. Moreover, data from the Wintermeyer group also indicate that interaction of the nascent chain with the uL23 inside the tunnel can also trigger changes in binding of SRP, another exit site ligand, which also interacts via uL23.

In Figure 2D, there is no control to show that SecA is essential in the strain being used (e.g. control strain lacking exogenous SecA with and without IPTG). The writing on the back of the plate is also quite intrusive. It would be essential to show the expression levels of the proteins relative to endogenous levels of SecA. It could be that the mutants are unstable in vivo and this causes the phenotype.

The functional analysis of the mutants would be more complete if their ability to support protein translocation (either in vivo or in vitro) was evaluated.

In Figure 3, there is a loss of two peptides from the MS analysis. Was the crosslinked peptide also detectable? This would strengthen the identification of the two peptides as the crosslinking site.

Was the auto-crosslinked complex tested for structural analysis? This might have stabilised the CTD allowing it to be visualised.

Does the ∆MBD construct with the 852 suppressor also form the auto crosslink? This would strengthen the argument that the locked ∆MBD conformation is a state the wild-type protein also encounters and not a non-physiological dead-end state.

In addition to 852, crosslinkers were also incorporated in the MBD at positions 893 and 898, but it was not clear why these were not further analysed. In particular, it would seem logical to text if these give rise to crosslinking to rProteins as that might permit further pinpoint localisation of the MBD at the ribosome surface or relative to the rest of SecA.

If Figure 5C, what is higher UV-specific band seen just above the SUMO-SecA band with the 209 construct? Is this a crosslink to another ribosomal protein? If so, that might further define the positioning of SecA relative to the ribosome.

In Figure 5D, the crosslink to the ∆MBD seems smaller. Is it established that this is still a crosslink to uL29 and not a different rProtein? This would be important to establish and confirm that ∆MBD binds in the same position as the WT, but cannot reposition upon binding the nascent chain.

In the authors model, peptide binding leads to release of SecA and the substrate from the ribosome. Can this be tested by releasing the nascent chain in the RNC.SecA complex with puromycin?

*Reviewer #3:*

The authors investigated the functions of SecA's C-terminal tail. Sedimentation and crosslinking assays with the purified motor protein and purified ribosomes or nascent chain ribosome complexes revealed that the metal binding domain binds to the ribosomal surface. The binding event causes the PPXD to shift to the open conformation, thereby destabilizing the flexible linker domain, which – according to small-scale angle scattering experiments – resides in the substrate protein binding groove. In turn, the autoinhibition is released and SecA is able to sample nascent polypeptides.

The manuscript offers new insight into the recognition process between SecA and secretory polypeptides. The approach appears to be sound. Yet both membrane and SecYEG are missing in the emerging picture. That is, SecA's substantial affinity to lipid membranes and SecYEG is not mentioned. How do the newly determined K_D_ values compare to those previously published for the interaction of SecA with lipid membranes and with the translocon? Does such affinity comparison favor SecA interactions with the nascent chain (i) in the cytoplasm or (ii) at the membrane surface?

[Editors’ note: what now follows is the editors’ decision letter after the first round of revisions.]

Thank you for submitting the revised manuscript entitled "The C-terminal tail of the bacterial translocation ATPase SecA modulates its activity" for consideration by *eLife*. Your revision has been carefully reviewed by a Reviewing Editor and a Senior Editor.

After carefully examining your responses to the editorial and reviewer comments, we find that the revision does not adequately address the concerns that were raised during review. Thus, the primary conclusions of this study are not supported with the level of rigor and depth expected for *eLife*. A detailed explanation of the concerns is provided below.

Editorial assessment of the author reply to reviewers:

The role of a metal requirement for MBD function was not examined satisfactorily. The sole experiment to address this issue, Figure 1D, is poorly controlled and uninterpretable. First, the signals in lane 7 and lane 2 are essentially indistinguishable (lane 7 is fainter, but the band is spread over a larger area, so the overall difference is hard to appreciate). Without replicates and statistics, one cannot reasonably infer relevance from such a modest difference. Even if one were to take the difference at face value, the effect cannot be attributed to the MBD from the information provided; it could possibly be due to an effect of EDTA on some part of the ribosome. There are no controls to verify that metal was in fact removed from the CTT. The authors have other experimental approaches at their disposal. For example, the purified CTT could be stripped of metal (e.g., with high concentration EDTA), the stripped metal and EDTA removed by dialysis or desalting, the extent of removal verified, and the metal-free CTT (compared to protein in which the metal was added back) used in the assays. Alternatively (or in addition), one could use mutants in the metal binding domain. In short, the issue of a metal requirement was not addressed.

The request to clarify the supposed UV-independent and highly efficient crosslinking from residue 852 was also not addressed adequately. This is a crucial experiment because it is the main direct evidence for the CTT engaging the catalytic core of SecA. The problem with UV-independence is that one cannot be certain the crosslink actually formed, with the alternative interpretation being a migration artefact with no crosslinks. The manuscript partially allays this concern, but the failure to detect a peptide is not especially strong evidence on its own. Based on what is shown, it could even be the case that BpA was not incorporated into 852 (amber suppression efficiency is known to be position-sensitive). While the authors could argue that the biotin blot shows amber suppression, this blot is very over-loaded and there is no negative control of a comparably overloaded SecA lacking the biotin tag. For these reasons, the conclusion that increased migration represents extremely efficient UV-independent intramolecular crosslinking through some unknown mechanism is not sufficiently compelling evidence to support a critical conclusion of the paper. If the authors' claim that this crosslink forms during the purification is correct, they can simply treat intact bacteria without or with UV, then harvest directly into SDS and monitor by blotting for the biotin tag. This should show a clear UV-dependent size shift to a position where the purified protein migrates. Other approaches are also feasible, including positioning BpA at nearby residues where the environment would be different, and hence, avoid constitutive UV-independent crosslinking. As it stands, a key conclusion of the study remains in doubt.

The request to provide the reader some estimate of the efficiency of binding in Figure 1D (first point of reviewer 2) was either ignored or misunderstood. The reviewer was asking simply to run 10% of the input sample on the same gel as the pulldowns so one could assess whether the amount pulled down was more or less than 10% of what went into the reaction. If it is far less, one could run 1% of the input (or whatever is roughly in the suitable range). The point is to provide the reader with a rough idea of how much is being pulled down in the experiment because this impacts how believable the findings are.

Similarly, the request to verify that SecA lacking the CTT fails to compete for binding in Figure 1D was also not addressed (second point of reviewer 2). This is a rather standard control, and it most certainly would affect the conclusions were it to compete similarly to WT SecA.

The reasonable request to test SecA levels in the different mutant strains relative to endogenous levels of SecA was addressed in a rather convoluted manner and did not adequately address the concern. IPTG-induced toxicity is not a good surrogate for expression levels because the basis of this toxicity could be different for different mutants (i.e., for some, it may have to do with SecA function, for others, it might have to do with protein aggregation or inappropriate interactions). It should be a straightforward matter to directly test expression levels by blotting using widely available SecA antibodies. Similarly, the request to directly test protein translocation is both reasonable and straightforward. The concern is not that SecA has some other function, but rather that ∆MBD is having its effect via a dominant toxicity unrelated to its function (which is certainly plausible).

The aberrant migration of the crosslink to ∆MBD relative to the corresponding crosslink to full length SecA in Figure 5D was not addressed. While it is true that ∆MBD migrates faster than wild type, the key concern was that the shift upon crosslinking was far smaller for ∆MBD than for full length, suggesting the possibility that the crosslinking partners are different. In this region of the gel, this difference in size shift is pretty substantial. It is therefore not acceptable to simply assume both crosslinks are to uL29.

[Editors’ note: what now follows is the decision letter after the authors submitted for further consideration.]

Thank you for submitting your article "The C-terminal tail of the bacterial translocation ATPase SecA modulates its activity" for consideration by *eLife*. Your article has been reviewed by three peer reviewers, and the evaluation has been overseen by a Reviewing Editor and John Kuriyan as the Senior Editor. The reviewers have opted to remain anonymous.

This revised article generally satisfies the reviewer comments. As you will see below, reviewer 2 feels you should be more cautious in some of your conclusions that are not fully convincing, particularly the intramolecular crosslink. In reading the article and evaluating the revisions myself, I have similar reservations.

Please adjust the text accordingly and submit a final version as per the instructions below.

*Reviewer #1:*

The manuscript is a new submission of a paper previously submitted to *eLife*, of which I was a reviewer.

Following review there were substantial (but not major) problems that needed addressing with the support of additional experiments and controls. It turned out that these could not be fully resolved within 2 months and the paper was returned to the authors.

I've now read the cover letter carefully and looked at the relevant sections of the new manuscript, and the JBC submission (available at bioRxiv) concerning the metal binding site. The new data addresses my initial query about the nature of the metal ligand, and consequences of binding.

It seems to me that all other concerns have also been addressed and would be happy to see the manuscript published.

*Reviewer #2:*

The authors have now addressed most of the major issues I had identified from the first round of reviews. They nicely show that a programmed short nascent chain does not block the crosslink to uL29 strengthening the authors model. They also show the 10% input for the binding assay in Figure 1, this reveals that the level of binding is relatively low, probably 1-2% of the input protein, i.e. binding to around 10-20% of ribosomes. They have also now attempted the suggested competition assay control (for original Figure 1D) but obtained inconclusive results with the ∆CTT construct. I would agree this compromises the interpretation of the competition experiment, hence its removal is sensible. It is puzzling why the assay was so variable considering that the WT and ∆CTT constructs apparently behaved well for the structural analysis.

Protein levels of the in vivo expression of the constructs is now shown and look similar, albeit without a loading control. I still also think it would have been informative to also look at the effects of the SecA mutants on translocation rather than growth alone.

The revised section of the text dealing with the internal crosslink now doesn't actually mention the auto-crosslink formation, save in the section- and figure headings. The section should explain the behaviour of the three constructs +/- UV and then introduce the uv-independent crosslink explanation. Also, as I mentioned previously, and was discussed by the other reviewers, the lack of identification of the crosslinked peptide tempers the definitive confirmation that the faster-migrating band is actually crosslinked. So this remains a slight weak point in the manuscript.

The altered mobility of the ∆MBD-uL29 crosslink is now addressed by the use of mass-spectrometry to confirm its identity.

*Reviewer #3:*

I have no additional comments to the revised manuscript.

---

## [Author Response]

[Editors’ note: the author responses to the first round of peer review follow.]

There are two main issues to be addressed experimentally:1) Experiments to determine whether and which divalent metal(s) impact function of the C-terminal domain. A complete analysis is probably beyond the scope of this paper, but initial experiments to examine the metal requirement were judged to be important.2) Disentangling the role of the nascent chain versus a programmed ribosome by analysis of RNCs containing short nascent chains.In addition, a number of clarifications, controls, and fuller explanations were requested (see the full reviews appended below). Each of these seem relatively straightforward. I suspect that for many of these, you will already have the answer from experiments not shown in the manuscript, although some might also require repeating experiments with some variations.Finally, it is worth addressing the issue of the interplay between the SecA-nascent chain interaction studied here as it relates to known interactions of SecA with lipids and SecYEG. Experimental investigation of this issue is beyond the scope of this study, but discussing this issue is worthwhile.

We have revised the manuscript in response to the comments by the reviewers. We feel that these revisions have significantly improved the manuscript, and we thank the reviewers for their very insightful feedback. We also have included a significant amount of new experimental material, which addresses the concerns raised in the decision letter. Our revisions include:

1) Crosslinking experiments indicating that SecA^Bpa399^ crosslinks to uL29 on RNCs containing arrested nascent SecM that is too short to interact with SecA (Figure 5—figure supplement 1B). These results suggest that the loss of crosslinking to uL29 in Figures 5D and Figure 5—figure supplement 1A was the result of interaction of SecA with the nascent chain (i.e. was not due to the state of SecM-stalled ribosome).

2) Ribosome binding experiments indicating that the presence of a metal chelator partially inhibits binding of the CTT to the ribosome (Figure 1D). These results suggest that binding of the MBD to its metal cofactor is important for efficient binding to the ribosome.

We did not include experiments addressing the identity of the physiological metal ligand. The identity of the metal cofactor does not affect our conclusions, and the apparent identity of the metal cofactor (zinc) is well established. Furthermore, although we have accumulated a significant amount of evidence suggesting that the physiological metal ligand is actually iron, we feel that challenging the established knowledge merits its own publication. Indeed, we are in the process of preparing this work for publication. A portion of this work is already publicly available as an author pre-print at BioRxiv (doi: 10.1101/173039), and we have discussed the possible implications of this in the revised Discussion section.

The reviewers also asked that we discuss the interplay between the SecA-nascent chain interaction and the already established interaction of SecA with SecYEG and lipids, and we have expanded our discussion to accommodate this request.

Reviewer #1:The manuscript submitted by the Huber lab relates to the regulation of the essential motor ATPase responsible for the bulk of bacterial protein secretion. The results build up to a hitherto unknown mechanism for its auto-inhibition and activation by nascent substrates at the ribosome exit site. The submission compiles high quality data, which have been sensibly interpreted, so the story is plausible. It is a story that should certainly be worth considering for those of us interested in SecA, and who have until now completely ignored the regulatory feature of the C-terminus. The paper will also be of interest to a much wider readership, and also for folk interested in the development of antibiotics. Thus, the work could be worthy of publication in eLife if the following points and suggestions can be addressed/considered.To be addressed:1) The metal binding site seems to have an important role here. Does it require metals to function? If so, which ones – Zn^2+^? Probably some/many of the experiments should be compared with and without divalent metals.

As noted in our Introduction, previous work suggests that the MBD binds to zinc. Unpublished research by our groups suggest that the physiological ligand of the MBD could be iron. However, we are not challenging the notion that MBD binds to zinc in the present work. Furthermore, both zinc and iron (as well as other metals) can promote the interaction of SecA with SecB. Determining the effect of any particular metal on the affinity of the MBD for the ribosome is extremely challenging. Ribosomes tend to contain large amounts of transition metal ions (including both zinc and iron). In addition, many cysteine-based iron-binding motifs will also bind to zinc with a similar or higher affinity. Because of the difficulty of these experiments and because the identity of the metal did not affect our conclusions, we did not address this topic.

2) Figure 3A shows important evidence for interaction of flexible linker with the catalytic core by intra-molecular crosslinking of residue 852. The authors say the SecA-852 band is fully crosslinked, in contrast to 893 and 898, because of its higher apparent mobility. I would like the authors to redo this experiment, but with equal quantities of SecA, just to check that it is not a loading artefact. Also, why is the crosslink not induced by UV ? Were the uncrosslinked samples kept in the dark at all times?

We have repeated this experiment numerous times with different levels of loading. We have noted the reproducibility of this experiment in the revised manuscript. We occasionally see that the protein migrates as a double band at early steps in purification but always converts to the aberrantly migrating form in Figure 3A and B following subsequent purification steps. We speculate that the reason for the increased efficiency of crosslinking is that the chemical environment surrounding position 852 activates the Bpa (or lowers the energy required for photoactivation of the Bpa). However, although the chemical basis for this increased efficiency is not known, it does not affect our conclusions.

3) Concerning the X-ray structure. If the FLD influences the conformation of SecA, as stated, then it should be evident in the high-resolution structure.

Not necessarily. If our model is correct, the FLD-induced autoinhibited state is only semi-stable. Thus, packing of the protein into crystals (a non-physiological condition) could easily favour a non-autoinhibited conformation. Indeed, we carried out the SAXS experiments because we were concerned about this possibility. We have made this line of thought more apparent in the revised text.

How does the new structure compare to others determined without the FLD?

As noted in the Results section, our structure is very similar to the previously published crystal structure of full-length SecA from *E. coli* (PDB file: 2FSG). The notable differences are that our structure is symmetrical and that the PPXD is better resolved. We have noted these differences more explicitly in the revised text.

Did the full-length protein fail to crystallise?

We did not attempt to crystallise the full-length protein as full-length SecA from multiple species has already been crystallised (and published) multiple times. As noted in the Introduction, the CTT is not resolved in any of these structures.

This figure could do with some improvement by colouring the different domains and highlighting the location of the C-terminus and FLD. The two monomers of the dimers could also be distinguished somehow.

The protomeric subunits of the apparent dimer (i.e. the “monomers”) are coloured differentially as noted in the figure legend. We have revised the figure and legend to make this point more clear. In addition, we have marked the location of the most C-terminal residue that is still resolvable with an asterisk in the figure.

As mentioned in the Results section, the FLD is not resolved in this structure. Its location is not marked because it is not known.

Reviewer #2:The study from Jamshad and colleagues investigates the function of the C-terminal domain of SecA, a key factor in protein export in bacteria. The authors show that in E. coli the C-terminal domain of SecA, as well as binding SecB, can also bind to the ribosome. Loss of the entire CTT leads to no obvious growth defect and actually enhances ribosome binding, suggesting that it likely has a regulatory function. Indeed through crosslinking, structural and functional analyses a model emerges where the metal binding domain of the CTT acts to control substrate loading in response to ribosome binding. Overall, the experiments are largely well performed and the study provides new insight into SecA function. Following attention to the points below I would be supportive of publication.In Figure 1D the SUMU-CTT is in large excess over ribosomes, it would be important to see a 10% input to assess the efficiency of binding.

The purpose of the experiments depicted in Figure 1D and E was to demonstrate that the MBD binds the ribosome and that it most likely does so at a specific site. Under physiological conditions the MBD is always connected to the catalytic core. The affinity of this domain for the ribosome in the absence of the catalytic core is not entirely relevant and does not affect our conclusion that the MBD binds to the ribosome near the site where full-length SecA binds the ribosome.

For the purposes of review, we note that preliminary experiments suggest that the affinity of SUMO-CTT for the ribosome is relatively weak. Lowering the input concentration of SUMO-CTT to 10% decreases the amount of co-sedimenting SUMO below the detection limit for western blotting. Concentrating the protein to carry out the converse experiment (10-fold more SUMO-CTT) results in protein solubility issues.

It would also be informative to assess whether SecA lacking the CTT is able to inhibit binding.

This is a technically challenging experiment that could provide limited (albeit potentially interesting) insight into the structure of the SecA-ribosome complex. Moreover, the results would not affect our conclusions. Thus, it is clearly beyond the scope of this work.

In Figure 2B, the effect of the nascent chain is established by comparing empty 70S ribosomes with a stalled SecM RNC. This could reflect changes due to the nascent chain, but also the fact that the ribosomes are now programmed. This could be distinguished using a shorter nascent chain that has not yet emerged from the exit tunnel.

Because we have already published this experiment, we have highlighted this publication more explicitly in the revised manuscript. We have also addressed this concern in greater detail with new experimental results presented in Figure 5—figure supplement 1B.

Moreover, data from the Wintermeyer group also indicate that interaction of the nascent chain with the uL23 inside the tunnel can also trigger changes in binding of SRP, another exit site ligand, which also interacts via uL23.

We assume the reviewer refers to the publication by Bornemann et al. (Nat Struct Mol Biol. 15(5):494-9). This paper suggests that the SRP binds with high affinity to translating ribosomes containing a signal anchor that is still buried in the exit channel. The possibility of programmed ribosomes is a very intriguing, but there is very little evidence to support their existence. Because the ribosomal structure is thought to be relatively rigid, it is difficult to imagine how signals could transmitted from deep in the exit channel to the ribosomal surface. In addition, ribosome profiling experiments by the Bukau group (Nature. 536(7615):219-23) do not support the existence of the high-affinity SRP-ribosome complex proposed by Bornemann et al. in vivo. Finally, mechanistic models published in subsequent papers by the Wintermeyer group (e.g.Nucleic Acids Res. 45(20):11858-11866) appear to suggest that the SRP does not distinguish between ribosomes containing nascent signal anchor motifs buried in the exit channel and those that do not.

In Figure 2D, there is no control to show that SecA is essential in the strain being used (e.g. control strain lacking exogenous SecA with and without IPTG).

The IPTG-dependent growth of these strains was noted in the Materials and methods section. We have also noted the IPTG-dependent growth in the Results section of the revised manuscript to make this more clear. We did not depict this experiment since the results were an empty plate.

The writing on the back of the plate is also quite intrusive.

We apologise for the intrusiveness of the labels. Because we take digital photographs of our plates, these labels are present to record the identity of the strain in the image. We cannot write directly on the image without altering it. Thus, we have retained the writing.

It would be essential to show the expression levels of the proteins relative to endogenous levels of SecA. It could be that the mutants are unstable in vivo and this causes the phenotype.

IPTG-induced expression of SecA is toxic when produced from a plasmid, and SecAΔMBD (and SecAΔCTT) becomes toxic at a similar concentration of IPTG, suggesting it is just as stable as wild-type SecA. In addition, the growth of strains producing SecAΔMBD from the chromosome was similar at all IPTG concentrations tested (10 μM to 1 mM), indicating that the growth defect is not due to differences in the steady-state level of the proteins. These results were noted in the original manuscript, and we have highlighted the more explicitly in the revised manuscript.

The functional analysis of the mutants would be more complete if their ability to support protein translocation (either in vivo or in vitro) was evaluated.

We assume that the growth defect of the strains expression SecAΔMBD is due to a defect in protein translocation since the only known essential function of SecA is as a translocation ATPase. We agree it would be interesting to carry out a full analysis of different Sec substrate proteins to determine which are affected. However, this is clearly beyond the scope of the current work.

In Figure 3, there is a loss of two peptides from the MS analysis. Was the crosslinked peptide also detectable? This would strengthen the identification of the two peptides as the crosslinking site.We spent months trying to identify a mass peak consistent with the crosslinked peptides to no avail. It is worth noting that the mass of the crosslinked peptide is *very* large – even for a crosslinked peptide. Because signal tends to decrease with the size of the peptide, our inability to detect this mass peak was disappointing but not surprising. Other factors – such as inhibition of trypsin as a result of conformation of the crosslinked protein or simply because the crosslinked peptide just doesn’t “like to fly” – could also have contributed to the low signal.Was the auto-crosslinked complex tested for structural analysis? This might have stabilised the CTD allowing it to be visualised.

Yes, it is possible to purify the auto-crosslinked protein in limited amounts with a significant amount of effort, and preliminary experiments suggest it is possible to crystallise this protein. Although these preliminary results are promising, producing crystals of sufficient quality would require a substantial amount of effort. Moreover, it’s possible that the resulting structure may not provide significant insight into the structure of the autoinhibited state. Thus, this experiment is beyond the scope of the current work.

Does the ∆MBD construct with the 852 suppressor also form the auto crosslink? This would strengthen the argument that the locked ∆MBD conformation is a state the wild-type protein also encounters and not a non-physiological dead-end state.

Unfortunately, the amount of time, effort and resources required to construct the mutant, purify the Bpa-incorporated protein and analyse it by mass spectrometry make it impossible to complete in a timely fashion.

In addition to 852, crosslinkers were also incorporated in the MBD at positions 893 and 898, but it was not clear why these were not further analysed. In particular, it would seem logical to text if these give rise to crosslinking to rProteins as that might permit further pinpoint localisation of the MBD at the ribosome surface or relative to the rest of SecA.

We analysed all three of these SecA^Bpa^ variants (i) for their ability to form the internal crosslinks and (ii) for their ability to crosslink to the ribosome. (Indeed, we initially purified these proteins in an attempt to determine the location of the CTT-binding site on the ribosomal surface.) We did not include these results because, with the exception of the internal crosslink in SecA^Bpa852^, none of the mutants produced internal or SecA-ribosome crosslink. The potential reasons for the negative result are myriad.

If Figure 5C, what is higher UV-specific band seen just above the SUMO-SecA band with the 209 construct? Is this a crosslink to another ribosomal protein? If so, that might further define the positioning of SecA relative to the ribosome.

We were intrigued by this band for the same reasons as the reviewer. If the previous structural models are correct, position 299 should be well positioned to interact with uL24. The size of the adduct is rather larger than expected for a crosslink to a ribosomal protein (which tend to be relatively small), but crosslinks do tend to make proteins run aberrantly. We spent several months attempting to determine the identity of the crosslinking partner but could not.

In Figure 5D, the crosslink to the ∆MBD seems smaller. Is it established that this is still a crosslink to uL29 and not a different rProtein? This would be important to establish and confirm that ∆MBD binds in the same position as the WT, but cannot reposition upon binding the nascent chain.

SecAΔMBD consistently migrates faster than full-length SecA in SDS-PAGE (see Figure 2B). Although the explanation suggested by the reviewer could be correct, we assume the difference in the running properties of this protein (and crosslinking adducts) is due to the fact that SecAΔMBD is 21 amino acids shorter than full-length SecA.

In the authors model, peptide binding leads to release of SecA and the substrate from the ribosome. Can this be tested by releasing the nascent chain in the RNC.SecA complex with puromycin?

This is an intriguing implication of our model but investigating it would clearly be beyond the scope of the present work. For example, investigating the rate of release would require kinetic measurements. As far as we understand, puromycin-induced release of arrested nascent polypeptides is very slow (and thereforelikely much slower than the binding kinetics of SecA for nascent chain). Fluorescence-based approaches may provide a more viable approach. However, these would take months or years to develop and optimise.

Reviewer #3:The authors investigated the functions of SecA's C-terminal tail. Sedimentation and crosslinking assays with the purified motor protein and purified ribosomes or nascent chain ribosome complexes revealed that the metal binding domain binds to the ribosomal surface. The binding event causes the PPXD to shift to the open conformation, thereby destabilizing the flexible linker domain, which – according to small-scale angle scattering experiments – resides in the substrate protein binding groove. In turn, the autoinhibition is released and SecA is able to sample nascent polypeptides.The manuscript offers new insight into the recognition process between SecA and secretory polypeptides. The approach appears to be sound. Yet both membrane and SecYEG are missing in the emerging picture. That is, SecA's substantial affinity to lipid membranes and SecYEG is not mentioned. How do the newly determined K_D_ values compare to those previously published for the interaction of SecA with lipid membranes and with the translocon? Does such affinity comparison favor SecA interactions with the nascent chain (i) in the cytoplasm or (ii) at the membrane surface?

These are interesting questions, and we thank the reviewer for raising them. We have included additional discussion to address the interplay between these different forms of SecA.

[Editors' note: what now follows is the author responses to the editors’ decision letter after the first round of revisions.]

Editorial assessment of the author reply to reviewers:The role of a metal requirement for MBD function was not examined satisfactorily. The sole experiment to address this issue, Figure 1D, is poorly controlled and uninterpretable. First, the signals in lane 7 and lane 2 are essentially indistinguishable (lane 7 is fainter, but the band is spread over a larger area, so the overall difference is hard to appreciate). Without replicates and statistics, one cannot reasonably infer relevance from such a modest difference. Even if one were to take the difference at face value, the effect cannot be attributed to the MBD from the information provided; it could possibly be due to an effect of EDTA on some part of the ribosome. There are no controls to verify that metal was in fact removed from the CTT. The authors have other experimental approaches at their disposal. For example, the purified CTT could be stripped of metal (e.g., with high concentration EDTA), the stripped metal and EDTA removed by dialysis or desalting, the extent of removal verified, and the metal-free CTT (compared to protein in which the metal was added back) used in the assays. Alternatively (or in addition), one could use mutants in the metal binding domain. In short, the issue of a metal requirement was not addressed.

To address this issue, we have included new experimental results in Figure 1D in which we modified the metal-coordinating cysteines in SUMO-CTT with AMS. Ribosome co-sedimentation experiments indicate that AMS modification almost completely abolishes binding to the ribosome. These results indicate that the MBD must be correctly folded in order to bind to ribosomes. We have referenced these results in subsection “Binding of the CTT to the ribosome.”.

In addition, the original editor’s summary requested more detail about “which divalent metal(s) impact function of the C-terminal domain”. At the time of revision, we were in the process of preparing a manuscript to address this issue, which was not yet ready of submission. The results described in this manuscript strongly suggest that the physiological ligand of the MBD is iron. This manuscript is currently being revised for submission to the Journal of Biological Chemistry and is available as an author preprint through BioRxiv (doi: 10.1101/613315). We reference these new results and speculate on their significance in paragraph four of the Discussion section in the revised manuscript. In the resubmission, purified SecA, SUMO-CTT and SUMO-MBD are bound to zinc. The identity of the metal could affect the strength of the interaction with the ribosome. However, it clearly does not disrupt ribosome binding entirely. Furthermore, the Zn-bound form of SecA binds with high affinity to SecB and is functional in vitro. Finally, maintaining SecA in the iron-bound form is technically challenging for a large range of reasons (e.g. contaminating zinc in purification buffers, rapid oxidation of iron under standard experimental conditions, etc.).

The request to clarify the supposed UV-independent and highly efficient crosslinking from residue 852 was also not addressed adequately. This is a crucial experiment because it is the main direct evidence for the CTT engaging the catalytic core of SecA. The problem with UV-independence is that one cannot be certain the crosslink actually formed, with the alternative interpretation being a migration artefact with no crosslinks. The manuscript partially allays this concern, but the failure to detect a peptide is not especially strong evidence on its own. Based on what is shown, it could even be the case that BpA was not incorporated into 852 (amber suppression efficiency is known to be position-sensitive). While the authors could argue that the biotin blot shows amber suppression, this blot is very over-loaded and there is no negative control of a comparably overloaded SecA lacking the biotin tag. For these reasons, the conclusion that increased migration represents extremely efficient UV-independent intramolecular crosslinking through some unknown mechanism is not sufficiently compelling evidence to support a critical conclusion of the paper. If the authors' claim that this crosslink forms during the purification is correct, they can simply treat intact bacteria without or with UV, then harvest directly into SDS and monitor by blotting for the biotin tag. This should show a clear UV-dependent size shift to a position where the purified protein migrates. Other approaches are also feasible, including positioning BpA at nearby residues where the environment would be different, and hence, avoid constitutive UV-independent crosslinking. As it stands, a key conclusion of the study remains in doubt.

To address this issue, we present new evidence in Figure 4—figure supplement 1. In this figure, we purified SUMO-SecA-biotin and SUMO-SecA^Bpa852^-biotin via their C-termini directly from cell lysates using streptavidin-coated magnetic beads. The two most prominent bands in the C-terminally purified SUMO-SecA^Bpa852^-biotin migrated with molecular weights identical to full-length SecA-biotin and to the faster migrating species in Figure 4 (previously Figure 3), indicating that both species are full-length SUMO-SecA^Bpa852^-biotin. We have referenced these new results in subsection “Auto-crosslinking of the FLD in the substrate binding groove of SecA” of the revised text. Exposure of lysates to light did not significantly increase the amount of the lower (apparent) molecular weight species. However, this result is consistent with our model if (a) crosslinking is very efficient and (b) the different populations of SecA (autoinhibited and uninhibited) are stable. We also note that previous NMR studies (Gelis et al., 2007) and our SAXS results are consistent with the proposed site of crosslinking.

We have also included additional text to provide a clearer explanation for the high efficiency of auto-crosslinking of SUMO-SecA^Bpa852^-biotin (paragraph five in the Discussion section). To summarise: the efficiency of Bpa crosslinking is influenced by three factors: (i) the amount of time the benzophenone group of the Bpa is in contact with the target molecule, (ii) the chemical reactivity of Bpa toward the target molecule and (iii) the amount of time the benzophenone group stays in the activated state. First, the results of this study and others (Gelis et al., 2007) is consistent with the idea that the FLD is stably bound in the substrate protein-binding groove of SecA, which could result in a long-lived contact between position 852 and the binding pocket. Second, although benzophenone can, in theory, react with any C-H bond, in practice it reacts with different efficiencies toward different amino acid side chains (Lancia et al., 2014; Wittelsberger et al., 2006). Finally, environment (hydrophobicity, pH, etc.) influences the photo-reactive properties of many aromatic compounds. Indeed, the excitation and emission spectra of 4-hydroxy-benzophenone, which has the same photo-reactive group as Bpa, are strongly affected by hydrophobicity and pH (Barsotti et al., 2015; Barsotti et al., 2017). These environmental conditions are normally assumed to be negligible because Bpa is normally incorporated at surface-exposed positions because in order to capture protein-ligand interactions. However, because our results indicate that position 852 is buried in a hydrophobic groove, the environment surrounding the Bpa could affect the photo-reactive properties of the benzophenone in unexpected ways. Any (or all three) of these factors could have influenced the efficiency of auto-crosslinking in this experiment. We apologise for the unclear explanation for the high efficiency of crosslinking in our previous reply.

The request to provide the reader some estimate of the efficiency of binding in Figure 1D (first point of reviewer 2) was either ignored or misunderstood. The reviewer was asking simply to run 10% of the input sample on the same gel as the pulldowns so one could assess whether the amount pulled down was more or less than 10% of what went into the reaction. If it is far less, one could run 1% of the input (or whatever is roughly in the suitable range). The point is to provide the reader with a rough idea of how much is being pulled down in the experiment because this impacts how believable the findings are.

We apologise. We completely misunderstood what the reviewer was requesting. We have addressed this issue by including 10% loading controls in the new experimental results depicted in Figure 1D. These results suggests that <20% of the ribosomes in the binding reaction are bound by SUMO-CTT in binding reactions containing 1 μM ribosome and 10 μM SUMO-CTT. Although this result suggests that the affinity of the MBD alone for the ribosome is relatively low, it is consistent with the decrease small decrease in the affinity of SecAΔCTT for the ribosome and it is consistent with the idea that the MBD normally binds to the ribosome in conjunction with the catalytic core. We have discussed have elaborated on these issues more extensively in paragraph two of the Discussion section.

Similarly, the request to verify that SecA lacking the CTT fails to compete for binding in Figure 1D was also not addressed (second point of reviewer 2). This is a rather standard control, and it most certainly would affect the conclusions were it to compete similarly to WT SecA.

As requested, we have repeated the binding competition experiment, but the results of this experiment were variable. The reason for the limited reproducibility is unclear. For example, it is possible that there is some experiment-to-experiment variability in the oxidation state of the metal-coordinating cysteines or in the bound metal cofactor in the MBDs of in SUMO-CTT and full-length SecA. However, the results presented in Figure 1 (and supplements) indicate that: (i) the MBD is well conserved (if not universally); (ii) the pattern of conservation suggests a binding partner besides SecB; (iii) the MBD binds to the ribosome; and (iv) the MBD binds to a site near uL23. Because it is possible to make these conclusions without the binding competition experiments, we have removed these results from the resubmitted manuscript.

The reasonable request to test SecA levels in the different mutant strains relative to endogenous levels of SecA was addressed in a rather convoluted manner and did not adequately address the concern. IPTG-induced toxicity is not a good surrogate for expression levels because the basis of this toxicity could be different for different mutants (i.e., for some, it may have to do with SecA function, for others, it might have to do with protein aggregation or inappropriate interactions). It should be a straightforward matter to directly test expression levels by blotting using widely available SecA antibodies. Similarly, the request to directly test protein translocation is both reasonable and straightforward. The concern is not that SecA has some other function, but rather that ∆MBD is having its effect via a dominant toxicity unrelated to its function (which is certainly plausible).

To address this issue, we present new results in Figure 2—figure supplement 3, which indicate that SecA is produced at similar levels in the three strains depicted in Figure 2D. We have referenced these results in the revised manuscript.

The aberrant migration of the crosslink to ∆MBD relative to the corresponding crosslink to full length SecA in Figure 5D was not addressed. While it is true that ∆MBD migrates faster than wild type, the key concern was that the shift upon crosslinking was far smaller for ∆MBD than for full length, suggesting the possibility that the crosslinking partners are different. In this region of the gel, this difference in size shift is pretty substantial. It is therefore not acceptable to simply assume both crosslinks are to uL29.

To address this concern, we analysed the crosslinking adduct produced by SecA∆MBD^Bpa399^ using LC-MS/MS. This analysis indicated that SecA∆MBD^Bpa399^ crosslinks to uL29. We have included this data in subsection “Site-specific crosslinking of SecA to ribosomes” of the revised text and in the legend for Figure 3.

[Editors' note: the author responses to the re-review follow.]

This revised article generally satisfies the reviewer comments. As you will see below, reviewer 2 feels you should be more cautious in some of your conclusions that are not fully convincing, particularly the intramolecular crosslink. In reading the article and evaluating the revisions myself, I have similar reservations.Please adjust the text accordingly and submit a final version as per the instructions below.Reviewer #2:The authors have now addressed most of the major issues I had identified from the first round of reviews. They nicely show that a programmed short nascent chain does not block the crosslink to uL29 strengthening the authors model. They also show the 10% input for the binding assay in Figure 1, this reveals that the level of binding is relatively low, probably 1-2% of the input protein, i.e. binding to around 10-20% of ribosomes. They have also now attempted the suggested competition assay control (for original Figure 1D) but obtained inconclusive results with the ∆CTT construct. I would agree this compromises the interpretation of the competition experiment, hence its removal is sensible. It is puzzling why the assay was so variable considering that the WT and ∆CTT constructs apparently behaved well for the structural analysis.

We agree that the lack of reproducibility is puzzling given how reproducible the experiment was in the past. SUMO-CTT has a very low affinity for the ribosome and because it is prone to misfolding due to oxidation or sequestration of the bound metal. Thus, some level of variability due to small differences in the fraction of active protein between batches is expected.

Protein levels of the in vivo expression of the constructs is now shown and look similar, albeit without a loading control. I still also think it would have been informative to also look at the effects of the SecA mutants on translocation rather than growth alone.

We thank the reviewer for noticing this. In addition to ensuring that we loaded samples with the same cell density, we also blotted against thioredoxin-1 as a loading control. However, we inadvertently forgot to include this loading control in the figure. We have revised Figure 2—figure supplement 3 to include the loading control.

The revised section of the text dealing with the internal crosslink now doesn't actually mention the auto-crosslink formation, save in the section and figure headings.

We did mention the auto-crosslink in this paragraph. However, our reference to it was not very clearly written. We apologise for the resulting confusion. We have now more explicitly mentioned this in the revised subsection (subsection “Auto-crosslinking of the FLD in the substrate binding groove of SecA”).

The section should explain the behaviour of the three constructs +/- UV and then introduce the uv-independent crosslink explanation.

The reason for the highly efficient intramolecular crosslinks is unknown. We speculate on several possibilities in the Discussion section. However, we agree that this could be confusing to the reader. We have therefore explicitly referred the reader to the Discussion where the possibilities are in depth.

Also, as I mentioned previously, and was discussed by the other reviewers, the lack of identification of the crosslinked peptide tempers the definitive confirmation that the faster-migrating band is actually crosslinked. So this remains a slight weak point in the manuscript.

We understand the reviewer’s (and the editor’s) concern. We have tempered our conclusions in the Results section (subsection “Auto-crosslinking of the FLD in the substrate binding groove of SecA”) and our interpretation in the Discussion (paragraph one and three) to address this weakness.